PLOS · Genetics

# Characterization of eclosion hormone receptor function reveals differential hormonal control of ecdysis during *Drosophila* development

Valeria Silva[1¤a], Robert Scott[2], Paulina Guajardo[1], Haojiang Luan[2], Ruben Herzog[3¤b], Benjamin H. White[2], John Ewer[1]*

**1** Instituto de Neurociencias, and Centro Interdisciplinario de Neurociencia de Valparaíso, Universidad de Valparaíso, Valparaíso, Chile, **2** Laboratory of Molecular Biology, National Institute of Mental Health, NIH, Bethesda, Maryland, United States of America, **3** Université de la Sorbonne, Institut du Cerveau - Paris Brain Institute - ICM, Inserm, CNRS, Paris, France

¤a Current address: Centre for Neural Circuits and Behaviour, The University of Oxford, Oxford, UK
¤b Current address: Instituto de Física Interdisciplinar y Sistemas Complejos (IFISC, UIB-CSIC), Campus UIB, Palma de Mallorca, Spain
* john.ewer@uv.cl

## Abstract

Neuromodulators and peptide hormones play important roles in regulating animal behavior. A well-studied example is ecdysis, which is used by insects to shed their exoskeleton at the end of each molt. Ecdysis is initiated by Ecdysis Triggering Hormone (ETH) and Eclosion Hormone (EH), which interact via positive feedback to coordinate the sequence of behavioral and physiological changes that cause exoskeleton shedding. Whereas the cell types targeted by ETH are well characterized, those targeted by EH have remained largely unknown due to limited characterization of the EH receptor (EHR). A gene encoding an EHR has been described in the oriental fruit fly, *B. dorsalis*, and in the desert locust, *Schistocerca gregaria*. However, little is known in these species about its expression pattern and its precise role at ecdysis, and no other insect EHRs are known. Here we analyze CG10738, the *Drosophila* ortholog of the *B. dorsalis* gene encoding EHR, and show that expressing it in cells confers sensitivity to EH. In addition, mutations of CG10738 specifically disrupt ecdysis, phenocopying the knockout of the EH gene. Together, these results indicate that CG10738 encodes the *Drosophila* EHR. As in *B. dorsalis*, EHR is expressed in the ETH-producing Inka cells; in addition, it is expressed in many known targets of ETH, including the neurons responsible for the secretion of other ecdysis-related peptides, such as CCAP and EH itself. Our results from targeted knockdown and rescue experiments reveal that EHR is required for ecdysis in diverse cell types and that the role of EHR in different targets differs with developmental stage. Our findings indicate extensive convergence of EH and ETH signaling and provide an exemplar of the complex mechanisms by which hormones control animal behavior.

**Data availability statement:** All data are in the manuscript and/or supporting information files

**Funding:** This study was funded in part by CONICYT (Comisión Nacional de Investigación Científica y Tecnológica, Chile; https://www.conicyt.cl/), Graduate Fellowship 21191720 (to V.S.), FONDECYT (Fondecyt Fondo Nacional de Desarrollo Científico y Tecnológico, https://www.conicyt.cl/fondecyt/; Chile) Grant 1221270 (to J.E.) and the Intramural Research Program of the National Institute of Mental Health, https://www.nimh.nih.gov/research/research-conducted-at-nimh; USA), Grant ZIAMH002800 (to B.H.W.). The contributions of the NIH authors were made as part of their official duties as NIH federal employees, are in compliance with agency policy requirements, and are considered Works of the United States Government. However, the findings and conclusions presented in this paper are those of the author(s) and do not necessarily reflect the views of the NIH or the U.S. Department of Health and Human Services. The funders play no role in the study design, data collection and analysis, decision to publish, or preparation of the manuscript.

**Competing interests:** The authors have declared that no competing interests exist.

## Author summary

Hormones and neuromodulators are important regulators of animal behavior. In insects, one of the best studied behaviors influenced by hormones is ecdysis, which allows the animal to shed the remains of its exoskeleton at the end of each molt. Ecdysis is controlled by two key hormones: Ecdysis Triggering Hormone (ETH) and Eclosion Hormone (EH). Whereas most targets of ETH have been identified, those of EH have remained largely unknown due to the limited characterization of its receptor, EHR. Previous studies identified a gene encoding EHR in the oriental fruit fly, *Bactrocera dorsalis*, and in the desert locust, *Schistocerca gregaria,* but little is known about its expression pattern or its precise role. Here, we show that CG10738 encodes the *Drosophila* EHR. We found that EHR is expressed in ETH-producing Inka cells, in neurons that secrete other ecdysis-related peptides, in addition to other neuronal classes and non-neuronal cells. Targeted knockdown and rescue experiments revealed that EHR is essential for ecdysis in various cell types, and that its role can vary depending on the developmental stage. Our findings reveal that the role of EH at ecdysis is complex and provides insights into how hormones and neuromodulators regulate animal behavior.

## Introduction

Peptide hormones and neuromodulators are one of the most diverse groups of signaling molecules found in animals [1]. These small chains of amino acids have diverse functions both within and outside the central nervous system (CNS) in regulating a wide range of processes that include metabolism, stress responses, circadian rhythms, locomotion, sleep, feeding, learning, memory, and social interactions [2–10]. They also play a role in regulating developmental processes such as insect metamorphosis, growth, and reproduction [11]. Through their multifaceted roles, peptide hormones and neuromodulators have profound effects on the physiology and behavior of organisms, highlighting their significance as essential modulators of neural communication [12,13].

Determining the mechanisms by which peptides control behavior is challenging due to the complexity of their actions. Indeed, they operate at different levels, involving various signaling pathways and cellular processes. They can act rapidly or over prolonged periods, and they may exert localized or widespread actions, affecting the entire CNS, specific neurons, as well as targets outside the CNS [13–17]. The synergistic actions of multiple peptides in interconnected networks further complicates efforts to understand their complex roles in behavior and physiology.

Ecdysis provides a unique system in which to investigate the mechanism of peptide action and the complex neuromodulation of a stereotyped behavior. This essential sequence of behavioral and physiological events occurs in all arthropods, where it is used to shed the remains of the old exoskeleton at the completion of each molt,

and to then inflate, harden, and pigment the exoskeleton of the next stage. This innate behavior includes three distinct and highly stereotyped phases: pre-ecdysis, ecdysis, and post-ecdysis, each characterized by specific timing and coordinated movements [18,19]. Any disruption or alteration in these ecdysis phases can have severe consequences including death.

Ecdysis is controlled by a complex interaction of neuropeptides and peptide hormones acting in a sequential order on the CNS [18,20,21]. Our current understanding of the endocrine bases of ecdysis proposes that ecdysis is initiated by the release of Ecdysis Triggering Hormone (ETH) from epitracheal Inka cells and of Eclosion Hormone (EH) from the ventromedial neurons (Vm) in the brain. These two peptides trigger each other's release, establishing an endocrine positive feedback that causes the sudden and near-complete release of both peptides [22–24] and provides an unambiguous, all-or-nothing, endocrine signal that triggers ecdysis. ETH and EH then act within the CNS, and previous work has provided information on the identity of many of the neurons that express the ETH receptor (ETHR) as well as on their timing of activation [25,26]. Thus, ETH acts on neurons that express the neuropeptides, Crustacean Cardioactive Peptide (CCAP), Myoinhibitory Peptide (MIPs), Bursicon, FMRF-amide (FMRFa), and Leucokinin, as well as on a large number of other peptidergic and non-peptidergic targets [19,26,27]. Although it is known that the actions of EH are not limited to causing ETH release but that it also acts within as well as outside of the CNS to control ecdysis [28,29], the identity and function of the EH targets are not known.

Here, we focus on the role of EH at ecdysis by identifying and characterizing the neurons and cells that express the EH receptor (EHR) in *Drosophila*. First, we confirmed that EHR is encoded by the CG10738 gene, the ortholog of a gene in *B. dorsalis* previously identified as an EHR [30]. We then used the Trojan exon technique [31] to generate an EHR-GAL4 driver and Split Gal4 hemidrivers, which we used to determine the expression pattern of the receptor and the function of EH targets as well as their pattern of activation during ecdysis. As expected, EHR is expressed in Inka cells; in addition it is expressed in the Vm neurons, which produce the EH neurohormone, suggesting that EH may participate together with ETH in an endocrine feedback mechanism that regulates EH release. EHR is also expressed in CCAP neurons, as well as many other neurons that express ETHR, revealing a convergence of the EH and ETH signaling systems. We found that disabling EHR function, silencing, or killing all EHR-expressing cells, was lethal at larval ecdysis, phenocopying the deficits of *Eh* null mutants [28]. By contrast, in some cases silencing EHR expression in specific subsets of cells only affected pupal or adult ecdysis, suggesting the existence of stage-specific EH targets or of targets that are essential only to ecdysis to a particular stage. Finally, we found that restoring EHR function in subgroups of EHR-expressing cells rescued normal ecdysis only partially and in a stage-dependent manner, suggesting that normal ecdysis may require EHR expression in diverse cell types, whose relevance may vary across different molts. Our findings underscore the complexity of the neuroendocrine control of ecdysis and enhance our understanding of the mechanisms by which neuropeptides function and modulate animal behaviors.

## Results

### CG10738 gene function is required for EH-induced ETH release from Inka cells

Previous research in the Oriental fruit fly, *Bactrocera dorsalis*, provided strong evidence that a gene homologous to the *Drosophila* CG10738 gene encodes an EH receptor expressed in the Ecdysis Triggering Hormone (ETH)-producing Inka cells [30]. To determine whether this receptor guanylyl cyclase has EHR activity, we determined whether genetic lesions in CG10738 (see S1 Fig) affected the ability of synthetic EH to trigger the release of ETH from Inka cells *in vitro*, which are known EH targets [22]. As shown in Fig 1A and quantified in Fig 1E (see also S3 Table), we found that trachea from control (*w*^1118^) 1st instar larvae at the dVP stage (double Vertical Plate; approximately 25 min prior ecdysis to the 2nd instar; [32]) challenged with 1 nM of synthetic EH released ETH from Inka cells in amounts similar to the near complete depletion of ETH that is seen in control animals at the end of ecdysis (Fig 1C). By contrast, no such release was observed (Fig 1B) in trachea from larvae *trans*-heterozygous for a genetic deletion, *Df(3L)Exel9017* (called here *Df(3)EHR*) which includes

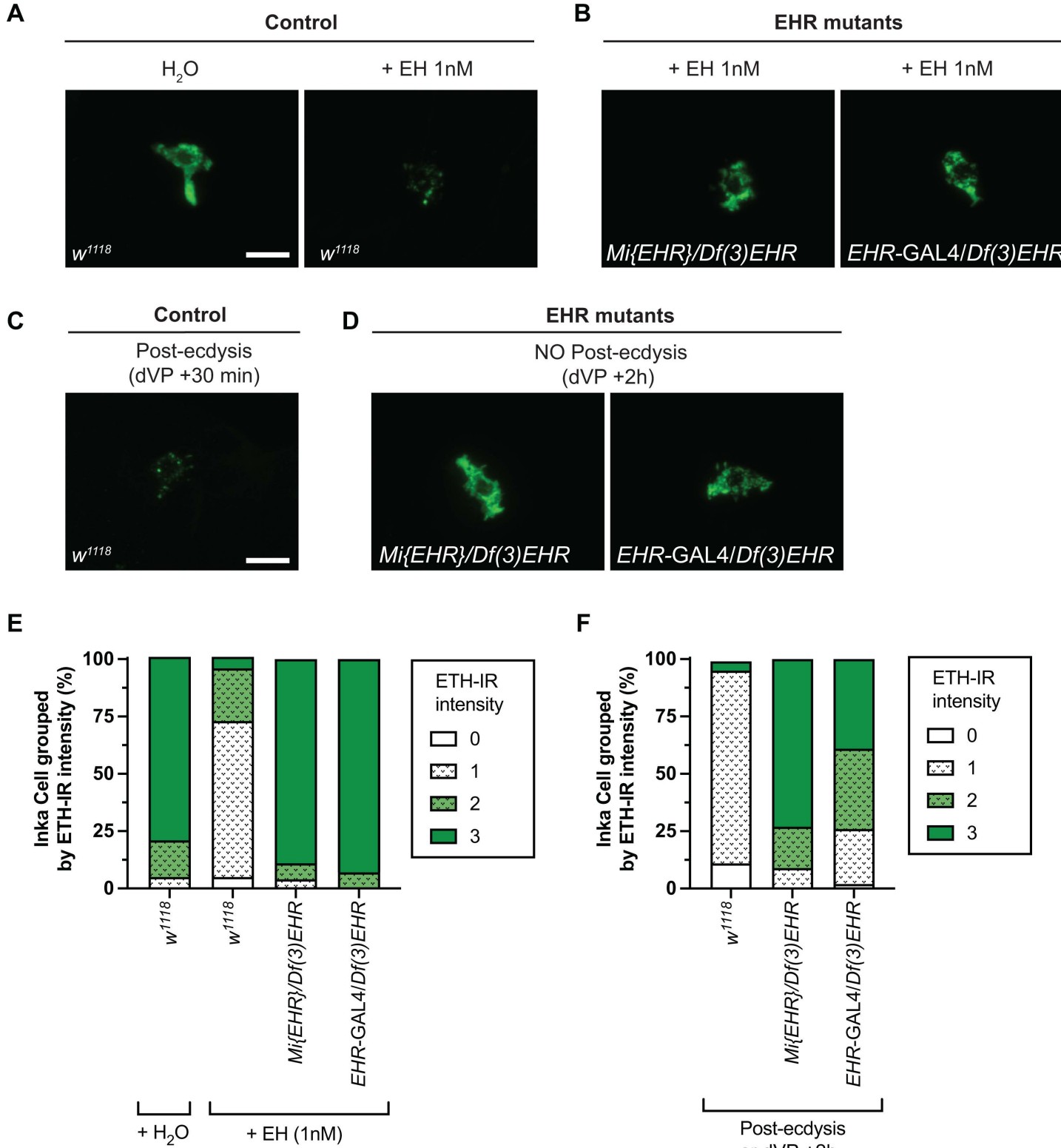

**Fig 1. Genetic lesions in CG10738 abolish EH-induced ETH release.** (A) Immunoreactivity to ETH (ETH-IR) of Inka cells from control animals (*w1118*) before (left) and after (right) *in vitro* stimulation with synthetic EH. (B) ETH-IR of Inka cells of EHR mutant animals, *Mi{EHR}/Df(3)EHR* (left) and

*EHR*-GAL4/*Df(3)EHR* (right), after *in vitro* stimulation with synthetic EH. (C) ETH-IR of Inka cells of wildtype (*w1118*) animals after ecdysis. (D) ETH-IR of Inka cells of EHR mutant animals, *Mi{EHR}*/*Df(3)EHR* (left) and *EHR*-GAL4/*Df(3)EHR* (right), 2h after dVP stage. (E, F) Quantification of ETH-IR intensity for experiments depicted in (A) and (B), and (C) and (D), respectively (N: 30–60 Inka cells, corresponding to 3–7 larvae per group). Animals that were less than 10 minutes before ecdysis were used for *in vitro* stimulation with synthetic EH (or solvent control); stimulating trachea from younger animals did not cause ETH release in wildtype animals. Results shown in panels E and F were analyzed using Fisher's exact test and show that the classes of intensities that are most different by inspection are statistically different (see results in S3 Table). Scale bar 10μm.

CG10738), and either a *Minos* insertion within exon 4 of CG10738 (called here *Mi{EHR}*) or a Trojan GAL4 driver in which GAL4 was inserted in the intron between exons 10 and 11, which was designed to interrupt the CG10738 gene (called here EHR-GAL4) (S1 Fig). A similar result was obtained in intact mutant animals examined 2h after dVP stage (Fig 1D and quantified in 1F and see also S3 Table).

## Mutations in EHR cause lethal phenotypes during ecdysis behavior

We next evaluated the behavior of animals *trans*-heterozygous for the *Minos* insertion in CG10738 or the GAL4 insertion in CG10738, and *Df(3L)Exel9017*. As summarized in Fig 2, these mutant animals expressed at ecdysis phenotypes similar to those observed in *Eh* null animals [28]. Indeed, they spent at least four times longer performing ecdysis-like behaviors compared to controls (*w1118*); they also did not express the pre-ecdysial phase of the behavior, and the majority failed to shed their old cuticle, often dying trapped within the old exoskeleton, and expressing the "buttoned-up" phenotype (Fig 2A and 2B and see also S3 Table) first described for ETH mutants [32]. Overall, approximately 80% of EHR mutant larvae failed to survive the first ecdysis (Fig 2E). Although most mutant larvae (~70%; Fig 2C) completely filled their trachea with air, they took approximately 25 min to do so in contrast to approximately 1 minute for control animals (Fig 2D). The ca. 20% of animals of either EHR mutant genotype that succeeded in shedding their old cuticle eventually all died during the 2nd to 3rd larval ecdysis or during the pupal ecdysis. Importantly, both the behavioral and the tracheal defects of these mutant combinations were substantially rescued by expressing CG10738 in the EHR-expressing cells using a Trojan exon GAL4 driver inserted in gene CG10738 (Fig 2 and S3 Table).

The lethal phenotype exhibited by CG10738 mutants during ecdysis (Fig 2) and the requirement of EHR function in ETH cells for the release of ETH upon stimulation with synthetic EH (Fig 1), provide direct evidence that CG10738 encodes the *Drosophila* EH receptor (EHR). The similarity of the EHR mutant phenotypes to those exhibited by animals lacking a functional EH gene further suggests that this is the only EH receptor in *Drosophila.*

## CG10738-expressing cells are necessary for larval, pupal, and adult, ecdysis

To determine the role of cells expressing CG10738 during larval, pupal, and adult ecdysis we first examined the consequences of inactivating or ablating all EHR-expressing cells using the inwardly rectifying potassium channel, *Kir2.1* [33], or the apoptotic inductor, *reaper* [34], respectively. In both cases we observed that virtually all larvae exhibited severe ecdysis deficits, including an extended locomotory phase, an absent or atypical pre-ecdysis phase, and an ecdysial phase that lasted at least 10 times longer than normal and was ultimately unsuccessful (Fig 3A and 3D and cf. S3 Table), with larvae dying trapped within their old cuticle with severe deficiencies in tracheal air filling (Fig 3B and 3C). We also used the TARGET system [35] to conditionally inactivate the CG10738-expressing cells 24h before pupal and adult ecdysis and found that this also caused a lethal phenotype during the corresponding ecdysis (Fig 3E and 3F). Similarly, activation all the EHR-expressing cells using the temperature-sensitive cation channel, *TrpA1*, for two hours prior to pupal or adult ecdysis caused 100% lethality (S2 Fig). These results demonstrate that CG10738-expressing cells are essential for larval, pupal, and adult ecdysis.

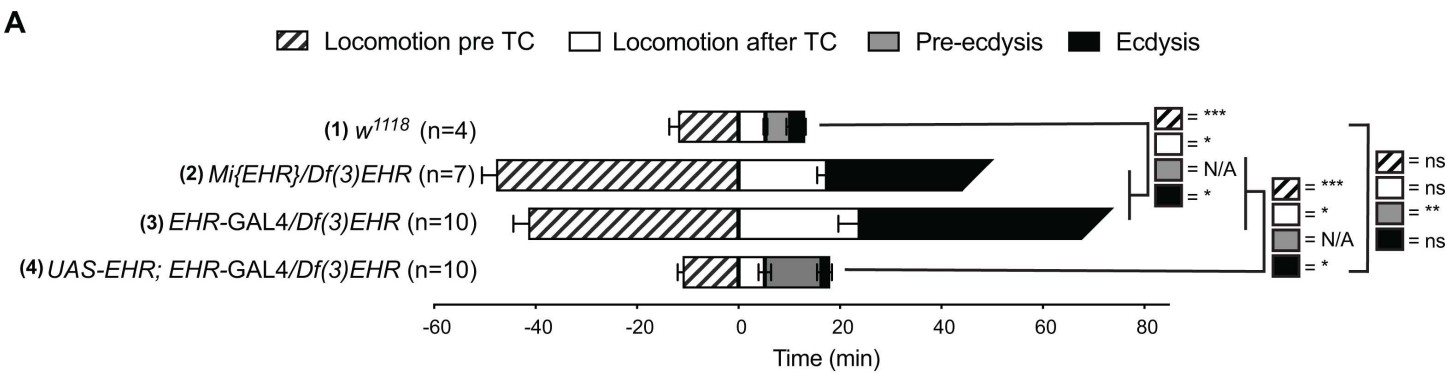

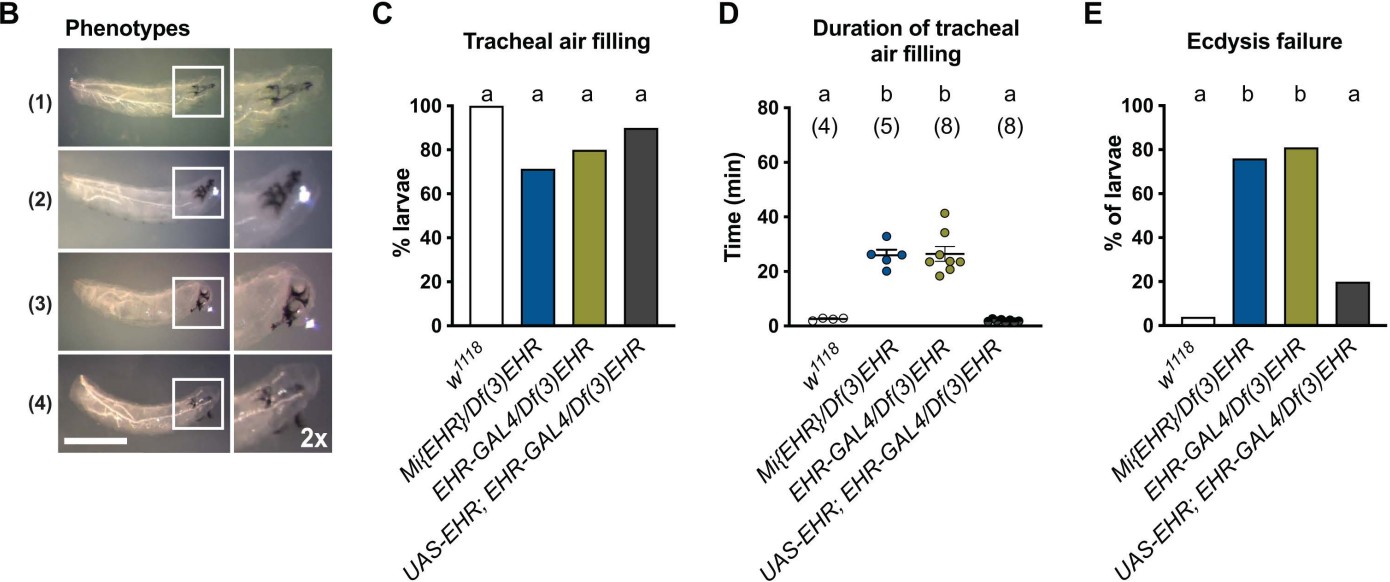

**Fig 2. EHR mutant animals fail at the first larval ecdysis.** (A) Duration of the different phases of L1 to L2 larval ecdysis. These phases include locomotion (crawling), before and after the collapse of the trachea (tracheal collapse, TC), pre-ecdysis, and ecdysis. Timeline has been aligned with respect to the time of TC. (B) Representative images of animals after ecdysis (for controls) or their terminal phenotype (for EHR mutants) for (1) Control (*w1118*), (2) *Mi{EHR}/Df(3)EHR*, (3) *EHR*-GAL4/Df(3)EHR, and (4) UAS-*EHR; EHR*-GAL4/Df(3)EHR (genetic rescue) animals. White frames indicate the region shown to the right with a 2x digital zoom. (C) Percentage of animals that completely filled their trachea with air. (D) Duration of tracheal air filling (for animals that completed this process; N in parentheses). (E) Percentage of animals that failed larval ecdysis (A-D, N = 4–10 larvae per condition; E, N = 21–31). Data were analyzed using an unpaired *t*-test (A), Fisher's exact test (C, E) and one-way ANOVA followed by Tukey's multiple comparison test (D). Statistically significant differences are indicated by *P ≤ 0.05, **P ≤ 0.01, *P ≤ 0.001, or ns for non-significant results in (A) and by letters in (C-E) (see exact values in S3 Table). Scale bar in (B) 500μm.

## EHR is expressed in ETH-producing Inka cells, in tracheal cells, in the CNS, as well as in other tissues, during all developmental stages

We used EHR-GAL4 to identify cellular targets of EH. Importantly, we found that EHR is expressed in Inka cells (Fig 4A-B), consistent with our previous results showing that EH-induced ETH release from these cells requires EHR function (Fig 1). EHR is also expressed in tracheal cells (Fig 4C and 4D), where it plays an essential role in the rapid inflation of the new tracheal tubes that occurs at ecdysis (Figs 2 and 3; [36,37]). We also observed that EHR is expressed throughout all developmental stages in some body wall cells (S3C and S3E Fig) and in the proboscis of the adult (S3E Fig). Finally,

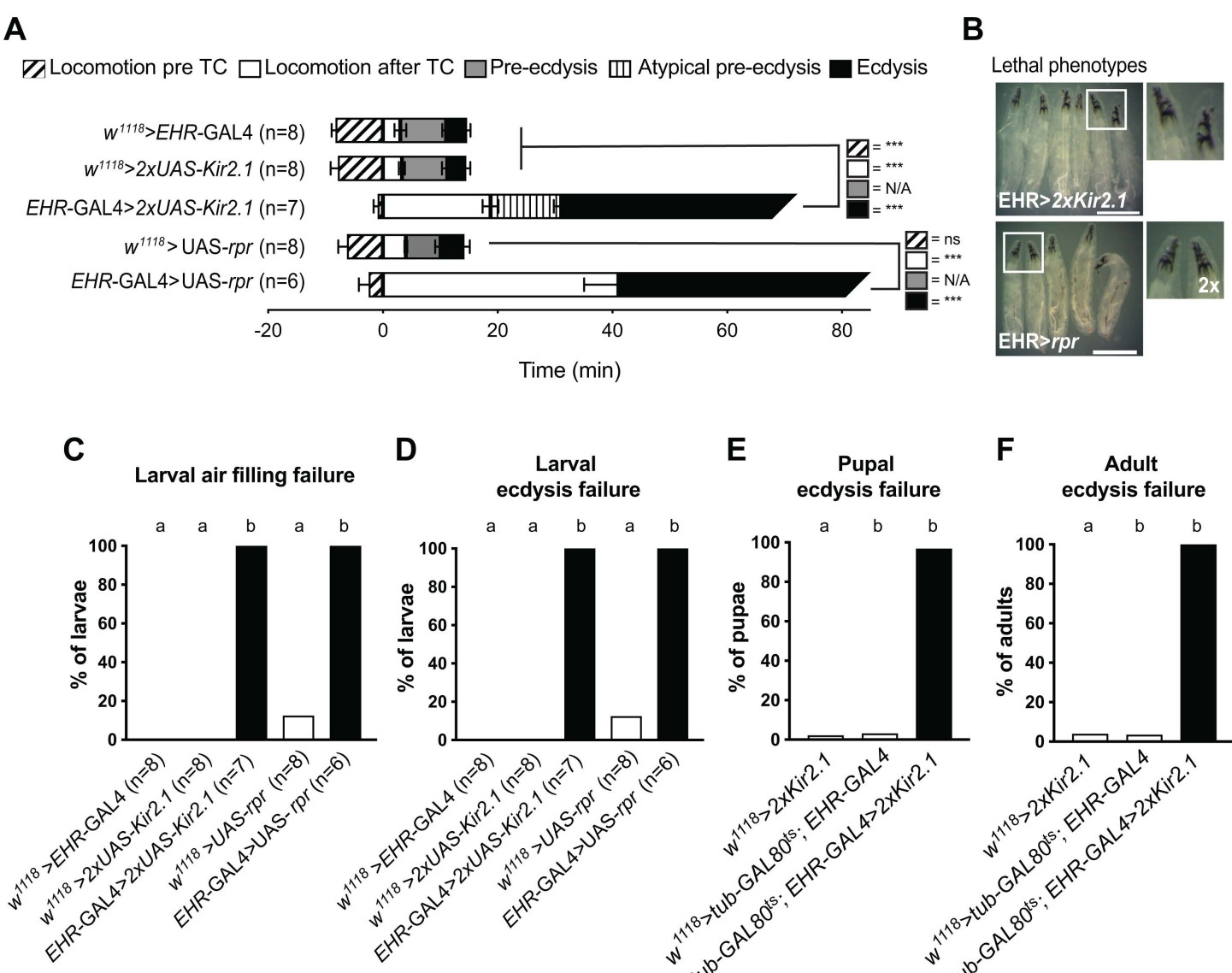

**Fig 3. Ablating or inactivating EHR-expressing cells causes failures at larval, pupal, and adult ecdysis. (A)** Duration of the different phases of L1 to L2 larval ecdysis; timeline has been aligned with respect to the time of tracheal collapse (TC). **(B)** Representative images of post-ecdysis or terminal phenotype of *EHR>2xKir2.1* and *EHR>rpr* animals. White frames indicate the region shown to the right with a digital 2x zoom. **(C)** Percentage of animals that failed to fill their trachea with air; (D-F) percentage of animals that failed larval **(D)**, pupal **(E)**, and adult **(F)**, ecdysis. (N: 150-500 per group for E-F). Data were analyzed using an unpaired *t-test* (A) or Fisher's exact test **(C-F)**. Statistically significant differences are indicated by *P ≤ 0.05, **P ≤ 0.01, *P ≤ 0.001, or ns for non-significant results in (A) and by letters in **(C-F)** (see exact values in S3 Table). Scale bar 500μm.

we observed EHR expression in leg imaginal discs (Fig 4E), which are associated with Keilin's organ. Interestingly, cells that ring Keilin's organ have been reported to express EH [29], which suggests a developmental signaling between this sensory structure and the developing leg.

We also found that EHR is broadly expressed in the larval (Fig 5) and pharate adult (Fig 6) CNS. In particular it is expressed in CCAP neurons at both stages (Figs 5B and 6B), which was also expected based on work from other insects [38]. By contrast, EHR expression in EH-producing Vm neurons (Figs 5C and 6C) was not expected, and suggests that

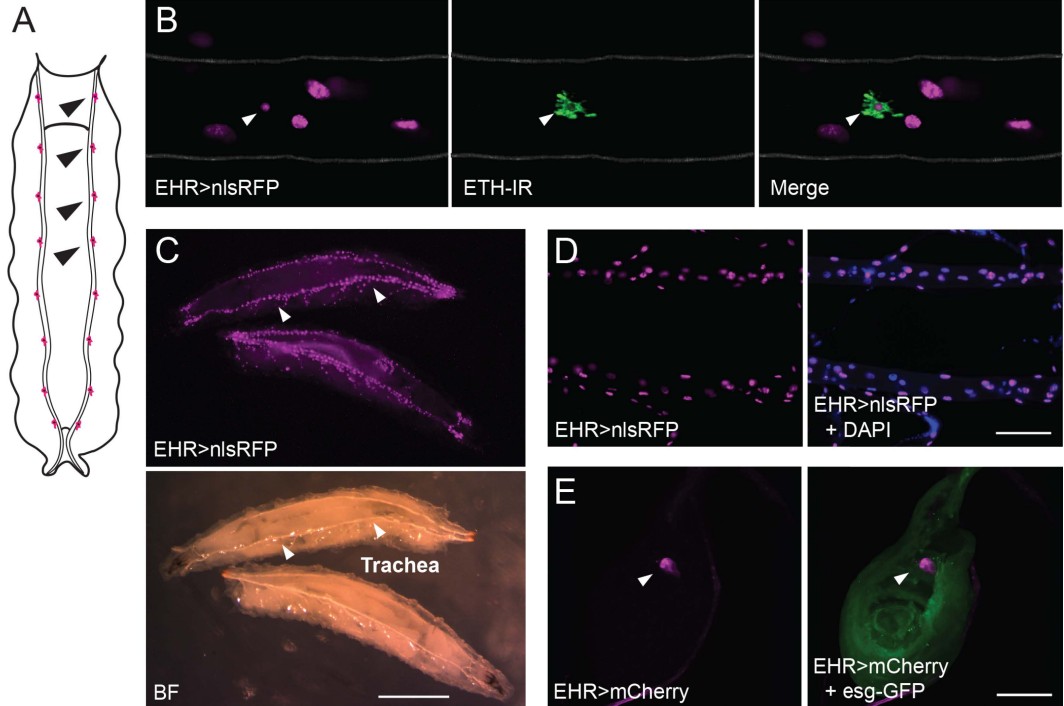

**Fig 4. EHR is expressed in the ETH Inka cells, tracheal epithelial cells, and the leg imaginal disc of the third instar larva.** (A) Schematic representation of a larva showing the Inka cells (colored in magenta; four are indicated by black arrowheads), which are located along the dorsal trunks of the trachea. (B) EHR expression in third larval instar Inka cell (white arrowhead) visualized using a nuclear localized RFP (*EHR*>nlsRFP, in magenta), together with ETH-immunoreactivity (in green). Grey lines indicate approximate borders of the trachea. (C) Top: prominent EHR expression in tracheal epithelial cells visualized in whole third instar larvae using a nuclear localized RFP (*EHR*>nlsRFP, in magenta); bottom: corresponding bright field image (BF). (D) Left: EHR expression in tracheal epithelial cells of third instar larvae using a nuclear localized RFP (*EHR*>nlsRFP, in magenta) and counterstained with DAPI (Right). (E) EHR expression in leg imaginal disc visualized using a nuclear localized RFP (*EHR*>nlsRFP, in magenta); disc is labeled using the *escargot* (*esg*)-GFP fusion protein (in green). Scale bars (D-E) 100μm, (C) 5mm.

the EH neuropeptide may participate together with ETH in controlling EH release. (By contrast, Inka cells do not express the ETH receptor (ETHR), S4 Fig.)

We then used Split Gal4 hemidrivers to further identify neurons that expressed EHR (for these drivers, GAL4 activity is reconstituted only in cells that express both a GAL4 DNA binding domain and a transcription activation domain, thereby making them intersectional tools). Using this tool, we further confirmed EHR expression in neurons expressing CCAP (Figs 5D, 6D and 6D') and in Vm neurons (Figs 5E, 6E, and 6E'; arrows), both of which also express the ETH receptor [26], as well as in the Dl EH neurons of the adult (Fig 6E; arrowheads). We also found that EHR is expressed in neurons that express the ETH receptor (ETHR, isoforms A and B) (Figs 5I-K, 6I, 6I'-K and 6K'), thereby revealing a notable level of convergence between the signaling of EH and ETH. In addition, we found that EHR is expressed in neurons that express the CCAP receptor (CCAP-R) (Figs 5F, 6F and 6F'), and the bursicon receptor, *rickets* (*rk*) (Figs 5H, 6G, 6H and 6H'), which are receptors of peptide hormones that act downstream of EH and ETH [17]. Bursicon itself is co-expressed in a subset of CCAP-expressing neurons, which, as expected, express EHR (Figs 5G, 6G and 6G'). These results reveal that EH may control ecdysis through both feed forward and feedback mechanisms. Finally, the use of split-GAL4 drivers showed that EH targets also include glutamatergic and cholinergic neurons (Figs 5L, 5M, 6L, 6L,'6M and 6M').

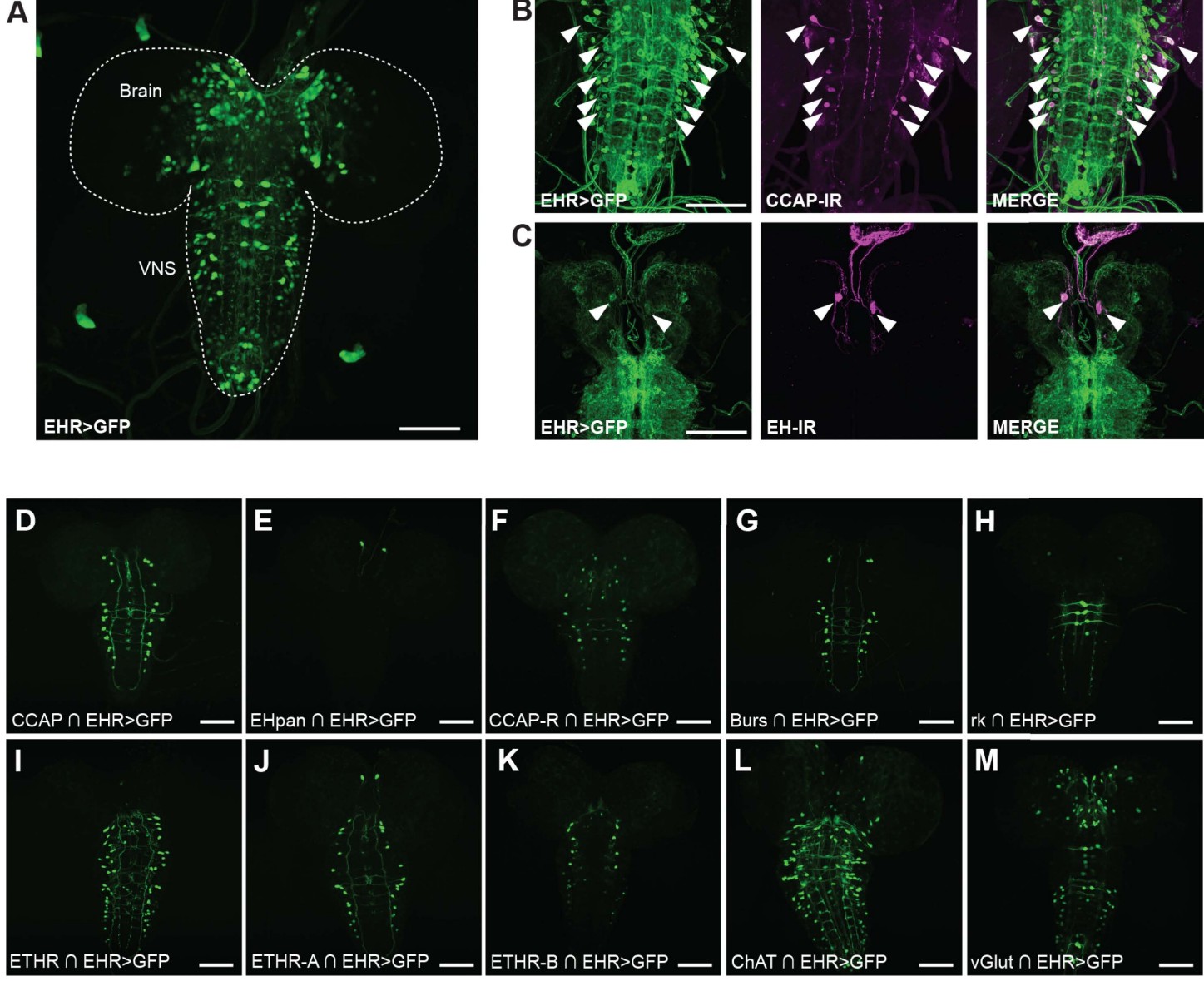

**Fig 5. EHR expression in the larval CNS.** EHR is expressed in a large number of neurons in the 3rd larval instar CNS (A). In particular, EHR is expressed in CCAP neurons (B, D) and in Vm (EH) neurons (C, E). It is also expressed in neurons that express the CCAP receptor (CCAP-R) (F); in bursicon (G) and bursicon receptor (*rickets*) neurons (H); in neurons expressing the ETH receptor (ETHR) (I) (both its A (J) and B (K) isoforms); and in cholinergic (ChAT) (L) and glutamatergic (VGlut) (M) neurons. In (A) EHR expression was visualized using *EHR>*GFP; in (B, C), colocalization was established by co-immunolabeling with anti-CCAP (B) and anti-EH (C) antibodies; in (D-M) different subgroups of EHR-expressing neurons were visualized using appropriate "split"-GAL4 hemidrivers (indicated by intersection symbol, "∩") together with UAS-GFP. Scale bar 100µm.

## Downregulating EHR in specific subsets of EHR-expressing cells causes lethal phenotypes

In order to determine the function during ecdysis of EHR in specific subsets of EH targets, we knocked-down EHR expression in different cell types. To do so we expressed EHR RNAi under the control of different GAL4 drivers (Table 1) and evaluated the effects on larval, pupal, and adult ecdysis. We found that knockdown of EHR with the broadly-expressed *tubulin*-GAL4 driver caused significant lethality at larval stages (failure at ecdysis from 1st to 2nd instar and 2nd to 3rd instar

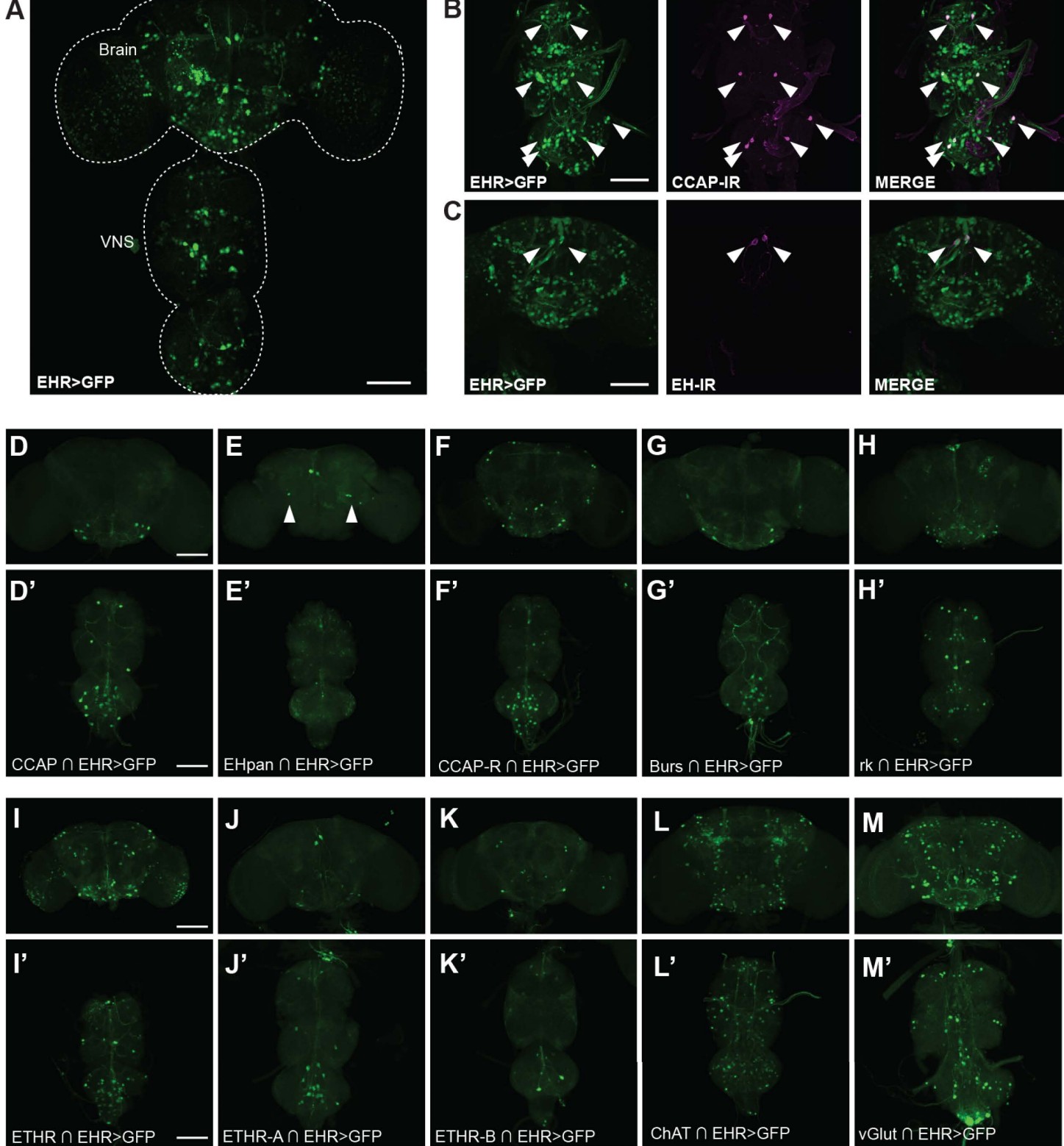

**Fig 6. EHR expression in the pharate adult *Drosophila* CNS.** EHR is expressed in a large number of neurons in the pharate adult CNS (A). In particular, EHR is expressed in CCAP neurons (B, D) and in EH neurons (Vm: C, E and Dl: E, arrowhead). It is also expressed in neurons that express

the CCAP receptor (CCAP-R) (F); in bursicon (G) and bursicon receptor (*rickets*) neurons (H); in neurons expressing the ETH receptor (ETHR) (I) (both its A (J) and B (K) isoforms); and in cholinergic (ChAT)(L) and glutamatergic (vGlut)(M) neurons. In (A) EHR expression was visualized using *EHR*>GFP; in (B, C), colocalization was established by co-immunolabeling with anti-CCAP (B) and anti-EH (C) antibodies; in (D-M) different subgroups of EHR-expressing neurons were visualized using appropriate "split"-GAL4 hemidrivers (indicated by intersection symbol, "∩") together with UAS-GFP. Scale bar 100μm.

was 47% and 95%, respectively), with the few (4) remaining 3rd instar larvae failing to ecdyse to the pupal stage. Overall, few of the spatially more restricted GAL4 drivers tested caused significant ecdysial defects when used to knockdown EHR expression (Table 1). Exceptions to this were GAL4 drivers expressed in peptidergic neurons (*C929*-GAL4 which drives expression in *dimmed*-positive peptidergic neurons; [39,40] and secretory cells (*386Y*-GAL4; [41]), the ETH cells and EH neurons, and tracheal cells (*btl*-GAL4), each of which yielded ecdysis deficits ≥20% at least at one stage. Interestingly, the impact of the knockdown was not uniform across developmental stages, and differences occurred even between the seemingly equivalent first and second larval ecdyses. Thus, for instance, knockdown of EHR in ETH cells caused most animals (69%) to fail at the second to third instar larval ecdysis, whereas no effect was observed at the first ecdysis. Similarly, knockdown of EHR in trachea significantly affected pupal and adult ecdysis, but did not affect larval ecdysis. Whether these results reveal stage-specific differences in the control of ecdysis or are simply the result of varying knock-down efficiency is unknown. (Knockdown efficiency may vary with driver strength, which itself may vary with developmental stage. Differences in driver strength likely account for such effects as EHups-GAL4 causing more severe phenotypes than the more widely expressed EHpan-GAL4 driver.) Interestingly, knockdown of EHR using *n-syb*-GAL4 (neuron-specific driver) and *386Y*-GAL4 had no effect on larval ecdysis but caused 100% failure at pupal ecdysis. This result suggests that larval ecdysis may also require EH actions outside of the CNS. In this regard, it is interesting to note that somatic sources of EH have previously been implicated in the control of larval ecdysis [29]. Expression of EHR RNAi in CCAP neurons did not block eclosion but caused wing expansion defects in over half of the animals, consistent with the fact that CCAP is not essential for ecdysis in *Drosophila*, but is co-expressed with bursicon, which controls wing inflation [42,43]. Interestingly, although there is substantial expression of EHR in ETHR positive neurons, we observed only modest defects in animals bearing knockdown of EHR in ETH targets, suggesting that EH may reinforce the response to ETH, but is not essential for ecdysial success. Finally, knockdown of EHR in CCAP-R- and *rk*- expressing cells had no overt effect on ecdysis. Thus, although the expression pattern of EHR suggests that EH may act on downstream elements of the ecdysial cascade, any such feedforward action appears not to be essential for successful ecdysis.

We were intrigued to observe EHR expression in body wall cells and in the proboscis (S3C and S3E Fig) because sensory neurons play a role at *Drosophila* pupation [44], and the cibarial pump is activated at adult emergence in the moth, *Manduca sexta* [45]. Interestingly, EHR knockdown using GAL4 drivers expressed in several different subgroups of sensory neurons (Ir25a, Ir7g, and 410-Gal4) impaired ecdysial success in more than 10% of animals at least one developmental stage.

## Peptidergic neurons expressing EHR are sufficient to rescue ecdysis behavior in EHR-mutants animals

In order to identify subsets of EHR-expressing cells that may play a critical role in the control of ecdysis behavior, we investigated whether expression of EHR in particular subsets of EH targets was sufficient to rescue the ecdysial failures caused by disabling EHR function. For this, we determined whether expressing EHR using different GAL4 drivers could rescue the defects of animals mutant for EHR (mutant animals used were *trans* heterozygous for *Df(3)EHR*, and either *Mi{EHR}* or *EHR*-GAL4). These results are summarized in Fig 7 and Table 2. As expected, expression of EHR using the EHR-GAL4 driver resulted in high levels of rescue (86–88% for larval ecdyses, and over 90% success rate in subsequent ecdyses). By contrast, most other drivers tested produced limited rescue. Overall, the most effective drivers were *C929*-GAL4 and *386Y-GAL4*, which are expressed in peptidergic or secretory cells, respectively; of these, only *C929*-GAL4

**Table 1. Percentage of ecdysial failure caused by downregulation of EHR in specific subsets of cells. Drivers causing frequent failures (20% or greater) are highlighted in orange.**

| GAL4 Driver | L1 to L2 | L2 to L3 | Pupal | Adult |
|---|---|---|---|---|
| w[1118] (control) | 0% (0/101) | 0% (0/73)[1] | 0% (0/72) | 0% (0/93) |
| **General** | | | | |
| tubulin | 47% (73/156) | 95% (79/83) | 100% (4/4) | N/A[2] |
| tsh | 0% (0/24) | 0% (0/48) | 4.5% (3/67) | 0% (0/64) |
| **Peptidergic cells** | | | | |
| c929 | 26% (7/27) | 29% (23/80) | 95% (53/56) | N/A |
| 386Y | 1.4% (1/73) | 0% (0/108) | 100% (122/122) | N/A |
| sNPF | ND[3] | ND | ND | 0% (0/94) |
| CCAP | 0% (0/25) | 0% (0/48) | 1.7% (2/121) | 0% (0/53)[4] |
| burs | 0% (0/85) | 0% (0/112) | 0.6% (1/175) | 1.1% (2/174) |
| EHups | 3.3% (1/30) | 39% (19/49) | 0% (0/27) | 0% (0/27) |
| Ehpan | 9.4% (6/64) | 3% (2/67) | 1% (1/94) | 4.4% (4/91) |
| ETH | 0% (0/118) | 69% (59/85) | 100% (87/87) | N/A |
| **Neuronal classes** | | | | |
| n-syb | 0% (0/19) | 0% (0/37) | 100% (146/146) | N/A |
| VGlut | 0% | 10% (13/130) | 0.9% (1/117) | 5.2% (6/116) |
| ChAT | 0% (0/41) | 0% (0/56) | 0% (0/112) | 0.9% (1/112) |
| DDC | 0% (0/26) | 0% (0/54) | 0% (0/46) | 0% (0/51) |
| ple | 0% (0/65) | 0% (0/58) | 0% (0/117) | 0% (0/151) |
| trhn | 0% (0/65) | 0% (0/62) | 0.6% (1/174) | 0% (0/188) |
| GAD1 | ND | ND | ND | 0% (0/30) |
| c164 | 7.1% (2/28) | 0% (0/38) | 6.4% (5/78) | 2.7% (2/73) |
| nompC | ND | ND | ND | 0% (0/74) |
| Gr66a | ND | ND | ND | 0% (0/146) |
| 109(2)80 | ND | ND | ND | 0% |
| Ir20a | ND | ND | ND | 0% |
| Ir8a | ND | ND | ND | 0% (0/56) |
| Smid c161 | ND | ND | ND | 0% (0/77) |
| Ir25a | ND | 20% (32/159) | 8.7% (11/127) | 5.2% (6/116) |
| Ir7g | ND | 14% (42/306) | 10% (26/253) | 0.8% (2/239) |
| Ir40a | ND | 6.9% (10/144) | 1.5% (2/134) | 3% (4/132) |
| 410 | ND | 15% (22/152) | 3.8% (5/130) | 4% (5/125) |
| **Ecdysis peptide receptors** | | | | |
| ETHR | 9.6% (27/280) | 13% (32/253) | 5% (11/221) | 0.4% (1/210) |
| CCAP-R | ND | ND | ND | 0% (0/23) |
| rk | ND | ND | ND | 0% (0/46) |
| **Other cells** | | | | |
| repo | 4.8% (2/42) | 0% (0/84) | 0% (0/94) | 7.1% (7/99) |
| btl | 4% (2/50) | 0% (0/52) | 54% (44/82) | 92% (35/38) |

[1]Fraction (and associated percentage) indicates the number of animals that failed at that ecdysis (in this case, fraction of 2nd instars that failed at ecdysis to the 3rd instar).

[2]No animals were obtained at this stage because an earlier ecdysis was lethal.

[3]Success at this ecdysis was not specifically examined.

[4]32/53 (60%) showed abnormal wing expansion.

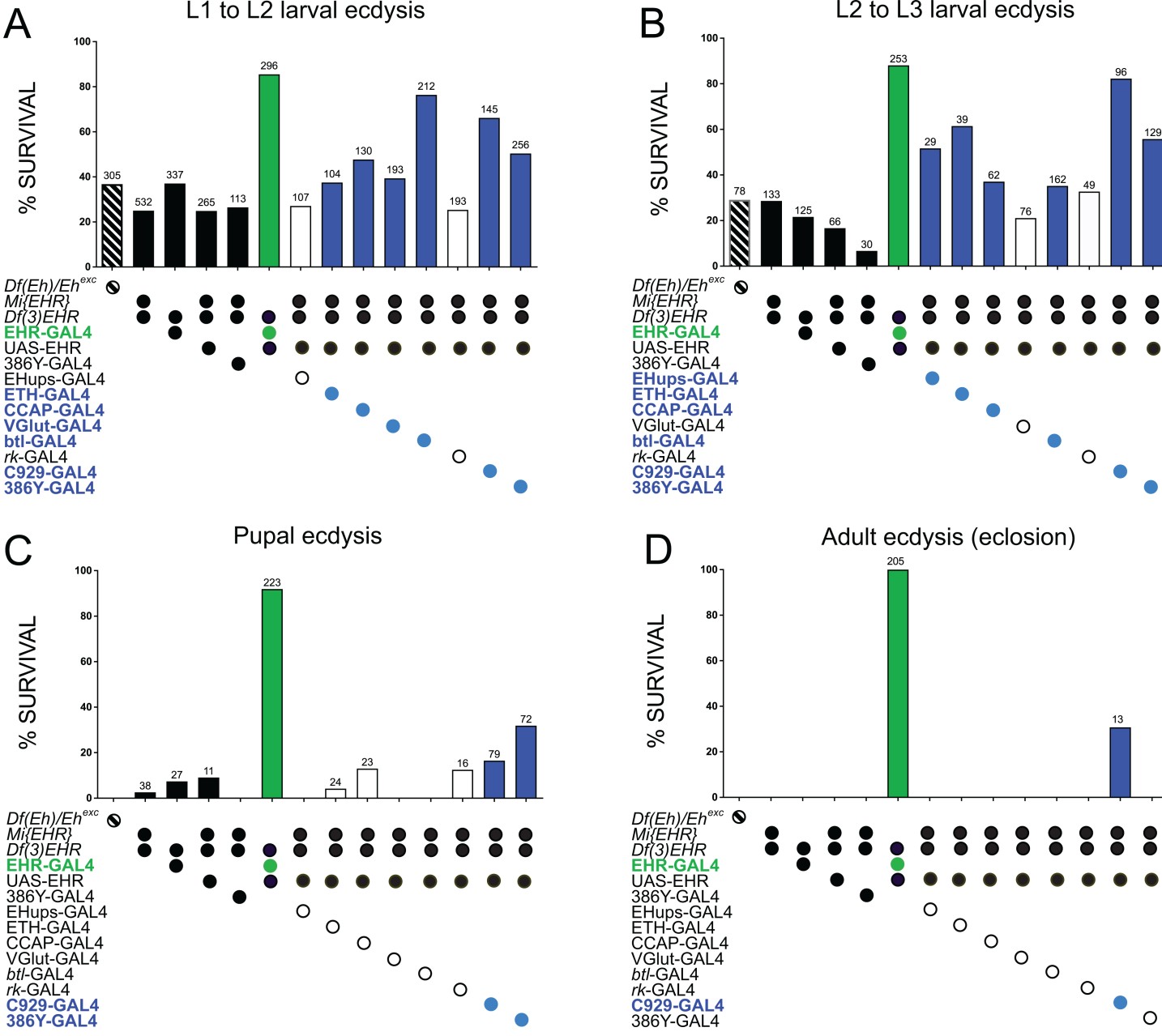

**Fig 7. Rescue of ecdysial defects obtained by expressing EHR in subsets of EH targets.** Percentage of successful ecdysis from the first to the second (A) and second to the third (B) larval ecdysis, and for pupal (C) and adult (D) ecdysis, when EHR was expressed under the control of the indicated GAL4 drivers in an EHR mutant background. Black bars summarize the phenotype of the different EHR mutants used; the green bars indicate the percentage of rescue obtained expressing EHR under the control of the *EHR*-GAL4 driver; the open and blue bars summarize, respectively, the results obtained for genotypes that did (blue bars) and did not (open bars) produce statistically significant levels of rescue compared to the average levels obtained for the various EHR mutants used. The genotypes that produced rescue are indicated in the corresponding color. Leftmost results summarize the results for an *Eh* null mutant combination (see [28] for a description of this genotype). Numbers above bars indicate the number of animals that completed that ecdysis. Values and percentages are also summarized in Table 2. Statistical analyses were performed using Fisher's exact test (see details in S3 Table).

**Table 2. Success of genetic rescue of EHR lethal phenotype obtained by expressing EHR using different GAL4 drivers.** These data are also summarized graphically in Fig 7. See S3 Table for results of statistical analyses.

| Genotype | L1 to L2 | L2 to L3 | Pupal | Eclosion |
|---|---|---|---|---|
| % Ecdysis Success in Mutant controls | | | | |
| Df(Eh)/Eh[exc] | 37%(305/829) | 29% (81/278) | 0% (0/78) | N/A[1] |
| Mi{EHR}/Df(3)EHR | 25% (133/532) | 29% (38/133)[2] | 2.6% (1/38) | (1/1)[3] |
| EHR-GAL4/Df(3)EHR | 37% (125/337) | 22% (27/125) | 7.4% (2/27) | 0% (0/2) |
| UAS-EHR; Mi{EHR}/Df(3)EHR | 25% (66/265) | 17% (11/66) | 9% (1/11) | (1/1)[3] |
| 386Y, Mi{EHR}/Df(3)EHR | 27% (30/113) | 6.6% (2/30) | 0% (0/2) | N/A[3] |
| % Ecdysis Success with Rescue of EHR function[4] | | | | |
| UAS-EHR; EHR-GAL4/Df(3)EHR | 86% (253/296) | 88% (223/253) | 92% (205/223) | 100% (205/205) |
| CCAP>EHR; Mi{EHR}/Df(3)EHR | 48% (62/130) | 37% (23/62) | 13% (3/23) | 0% (0/3) |
| ETH>EHR; Mi{EHR}/Df(3)EHR | 38% (39/104) | 62% (24/39) | 4.2% (1/24) | 0% (0/1) |
| EHups>EHR; Mi{EHR}/Df(3)EHR | 27% (29/107) | 52% (15/29) | 0% (0/15) | N/A |
| VGlut>EHR; Mi{EHR}/Df(3)EHR | 39% (76/193) | 21% (16/76) | 0% (0/16) | N/A |
| UAS-EHR; 386Y, Mi{EHR}/Df(3)EHR | 50% (129/256) | 56% (72/129) | 32% (23/72) | 0% (0/23) |
| C929>EHR; Mi{EHR}/Df(3)EHR | 66% (96/145) | 82% (79/96) | 17% (13/79) | 31% (4/13) |
| rk>EHR; Mi{EHR}/Df(3)EHR | 25% (49/193) | 33% (16/49) | 13% (2/16) | 0% (0/2) |
| btl>EHR; Mi{EHR}/Df(3)EHR | 76% (162/212) | 35% (57/162) | 0% (0/57) | N/A |

[1]Success at this ecdysis could not be evaluated because an earlier ecdysis was lethal.

[2]Fraction (and associated percentage) indicates the number of animals that ecdysed to the next stage (in this case, fraction of 2nd instars that ecdysed to the 3rd instar).

[3]Due to the low sample number, this outcome is not indicated as "100%" rescue.

[4]Cells are color coded as described in Fig 7.

rescued adult ecdysis, albeit only modestly. In addition, the results obtained were many times stage-dependent and were greater for larval than for pupal or adult ecdyses, again suggesting that some EH targets are preferentially involved in the control of some of the fly's ecdyses. A caveat is that just as the absolute levels of knockdown are difficult to evaluate due to the possible confounding effect of driver strength, the levels of rescue obtained using different GAL4s are also difficult to compare quantitatively.

### EHR-expressing neurons show a complex pattern of activity during fictive pupal ecdysis.

In order to visualize the pattern of activity induced in EH targets at ecdysis, we used synthetic ETH to stimulate the excised pupal CNS from animals expressing GCaMP6s under the control of the EHR driver (as described in [25]). Fig 8A-C and S1 Video shows the timecourse of activation of all individual EHR-expressing neurons after hormonal stimulation *in vitro*. We observed that most neurons were active only during the presumed ecdysial phase (min 10–30 of the recording), although a small group of neurons became active earlier (which might correspond to the pattern of activity of fictive pre-ecdysis) and others became active later, which might correspond to fictive post ecdysis. Thus, as has been observed previously [19,25], different targets of ecdysial peptides respond at different times following the simultaneous stimulation with ETH, revealing that a complex pattern of neuronal activity is set in motion following the sudden release of the triggering neuropeptides, ETH and EH.

### Early Activation and Elevated Functional Coupling of CCAP Cells in the EHR Population

In order to understand the temporal dynamics within the EHR-expressing cell population, we asked whether specific neuronal populations became active earlier and displayed stronger functional coupling to the overall pattern of activity.

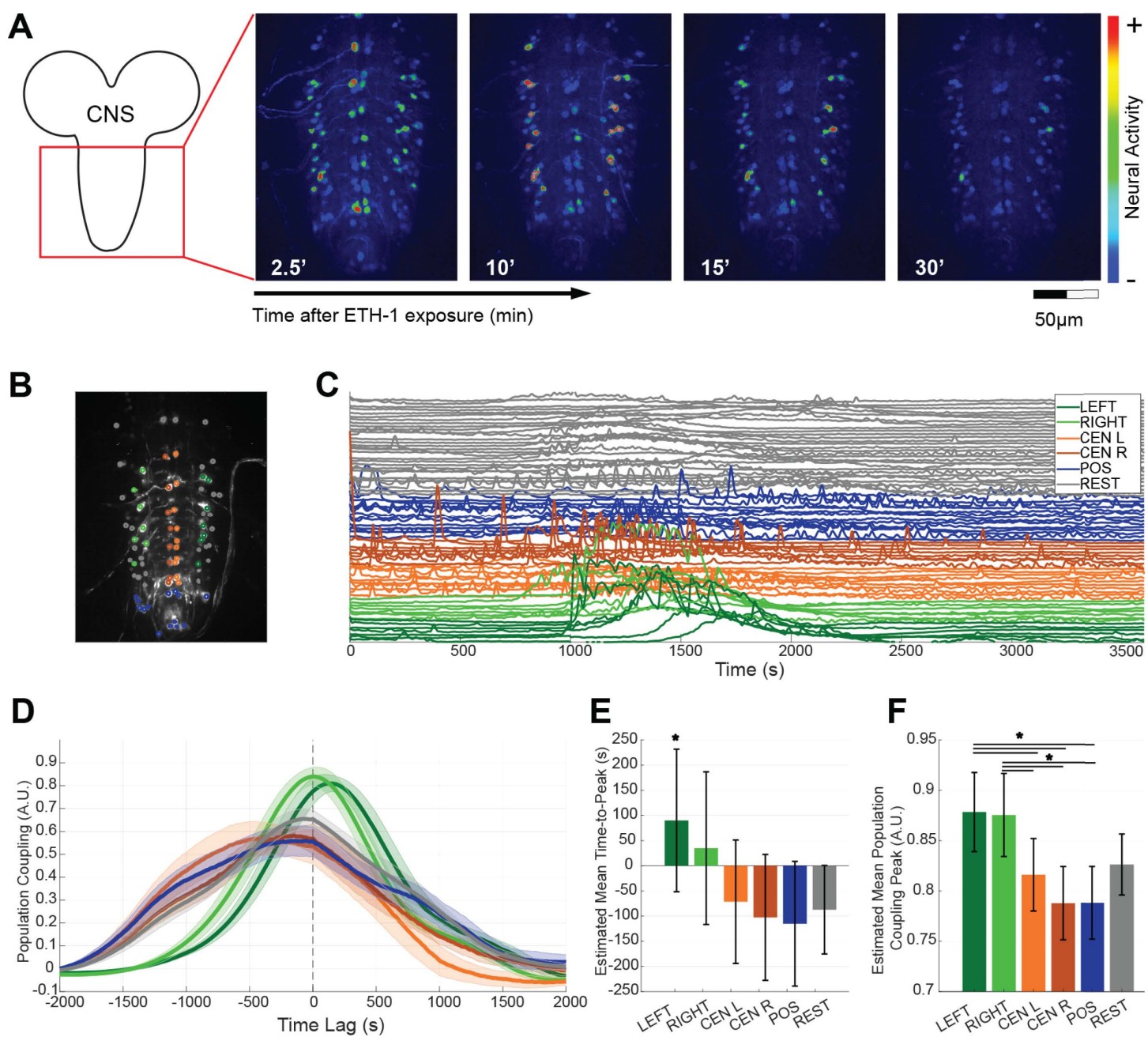

**Fig 8. Functional population analysis of the time course of activation of EHR neurons in response to ETH stimulation *ex vivo*.** (A) *Left:* Reference image of a pharate pupal CNS showing the area where the CNS response was recorded. *Right*: representative frames of GCaMP signal recorded *ex vivo* from CNS expressing EHR>GCaPM6s, captured at different times after ETH challenge. CNSs were imaged for 1h, with images captured at a frequency of 0.2Hz. (B-F) Functional population analysis. (B) CNS with different subpopulations of EHR neurons color coded based on their position: "Left" (dark green) and "Right" (light green), which are likely made up mostly of CCAP neurons; "Central Left" neurons (orange); "Central Right" neurons (dark orange); "Posterior" neurons ("POS", blue); and the remaining ("REST") neurons (gray). (C) Timeseries of GCaMP6s signal following the same color code used in B. Note the increased calcium activity around ~1000s (15 min) after stimulation with ETH (see also S1 Video). (D) Population coupling function for the 6 different cell types for the CNS shown in (B) and (C). Solid lines represents the average for all the cells of the same type and shaded area is the standard error. Positive lag values means that the population activates after a given cell. (E) Estimated mean time to peak of the functional population coupling function (PCF) from 5 different experiments. Asterisk denotes statistically significant differences (* p<0.05) estimated using a linear-mixed effect model. (F) Estimated mean peak of the PCF. Horizontal lines denote significant differences using Bonferroni test corrected Wilcoxon rank sum (* p<0.05) (see details in S3 Table). Estimations of means and confidence intervals were derived using a linear-mixed effect model with experiments as random effects. Error bars for (E) and (F) are 95% confidence intervals.

To this end, we defined six groups of neurons based on their position (Fig 8B; Left and Right, which are likely made up mostly of CCAP neurons; Central Left [CEN L], Central Right [CEN R], posterior neurons [POS] and the remaining neurons ['REST']; see Methods for further details and S1 Video) and computed their population coupling following Okun, *et al*. (2015) [46]. (We chose to focus on the CCAP neurons because they can easily be identified in such preparations.) For this, we calculated for each cell group and across 5 different videos, the cross-correlation function between its activity and the average activity of the rest of the population (Fig 8D). The temporal position (Fig 8E) indicated whether the cell became, on average, active before or after the rest of the population and the maximum of this function served as an index of population coupling strength (Fig 8F).

Our analysis demonstrated that CCAP cells, and in particular the "Left" subgroup, reached their peak coupling approximately 90 seconds prior to the activation of the rest of the network ($p < 0.05$, linear-mixed effect model with experiments as random effects). The "Right" subgroup showed the same trend, but without significant effect ($p = 0.17$). Furthermore, the amplitude of the positive peak was significantly higher for both Left and Right CCAP subpopulations compared to that of all other EHR-expressing populations (Bonferroni corrected Wilcoxon rank sums *$p < 0.05$).

These findings position CCAP cells as early activators within the EHR population, exhibiting significantly higher functional coupling that may serve to prime subsequent population-wide activity. The marked temporal precedence and enhanced coupling of CCAP cells suggest that they play a critical role in orchestrating coordinated network dynamics.

## Discussion

"Eclosion hormone" activity was described almost 50 years ago [47,48], and EH has since emerged as one of the main actors controlling ecdysis behavior in insects. However, the breadth of its action has remained largely unknown because of the limited characterization of its receptor. Here, we demonstrate that the gene CG10738 encodes the *Drosophila* EH receptor (EHR) and show that it is expressed in all the cells and neurons previously implicated in the control of ecdysis in addition to a large number of neurons whose identity is mostly unknown. Interestingly, EH targets include the principal EH-producing neurons, many ETH targets, as well as neurons that are targets of CCAP and of bursicon, which act downstream of EH and ETH. This pattern of expression suggests that EH and ETH control ecdysis via complex neuromodulatory feedback and feedforward loops. EH signaling to neurons that express CCAP and bursicon, as well as to downstream neurons targeted by these two peptides, demonstrates the existence of feedforward loop that overlaps with the one previously described for ETH [19,26]. The latter loop includes the EH-expressing Vm neurons and the CCAP/bursicon neurons, and our results here indicate the presence of other possible loops formed with additional neurons that are jointly targeted by EH and ETH. These data indicate that the neuromodulatory networks that control ecdysis include coupled feedforward loops organized in a hierarchical manner, consistent with the model proposed earlier by Diao et al. (2017) [17]. Sequential execution of ecdysis behavior is presumably determined by the progression of activity through these loops, which also incorporate feedback control, mediated at least in part by EH. In addition to the positive feedback loop formed by EH and ETH signaling, our results suggest an additional possible autocrine loop in which the Vm neurons respond to their own release of EH. However, some caution is required in interpreting this finding since it is possible that the Vms are responding to EH released by the small subset of recently discovered "non-canonical" EH-expressing neurons, which are not targets of ETH [29]. In general, however, the topology of neuromodulatory connections among ecdysis neurons suggests broadly distributed, coordinated control of behavior in which feedforward loops support staggered, progressive changes in motor program output.

The broad importance of EH signaling in the control of ecdysis is underscored by the phase- and stage-specific effects of manipulations of EHR function in subsets of EH targets. Thus, whereas some subsets were essential for the proper execution of all ecdyses, others were required for only pupal and adult ecdysis. Interestingly, we failed to find small subsets of EHR-expressing targets that were sufficient to rescue the ecdysial defects of most individuals of a given genotype suggesting that many of the EH targets are important for successful ecdysis. The proposal that EH action is distributed

over multiple targets is consistent with the broad expression of EHR, the fact that many EHR-expressing neurons are activated by EH *in vitro*, and that expression needs to be restored in much of the EHR pattern to obtain rescue. Nevertheless, our findings underscore the relevance for successful ecdysis of EHR expression in Inka cells, in trachea, and in *dimmed*-expressing peptidergic neurons (which are defined by the C929 GAL4 driver; [39,40]).

Although the majority of neuropeptide receptors belong to the G protein-coupled receptors (GPCRs) family [49], the EH receptor belongs to the family of guanylyl cyclase receptors (GCR). Previous research by Chang *et al.*, (2009) [30] characterized the GCR, BdmGC-1, from the Oriental fruit fly, *Bactrocera dorsalis*, and showed that it mediates robust increases of cGMP following exposure to EH, when expressed in HEK cells. They also showed that BdmGC-1 is strongly expressed in the Inka cells and that these cells respond with an increase in cGMP-immunoreactivity to an in vitro challenge with synthetic EH. Recently, Verbakel *et al.* (2024) [50] showed that knockdown of the locust ortholog of BdmGC-1 causes failures at ecdysis. Our results here confirm that the *Drosophila* ortholog of BdmGC-1, CG10738, encodes the *Drosophila* EH receptor and greatly extend what is known about EHR distribution and function.

Although EH is known to act within the *Drosophila* CNS [28], neuronal targets of EH had not been identified directly. In many insects, cGMP-immunoreactivity (cGMP-IR) increases at ecdysis in CCAP neurons [51,52] This is also true in some dipterans, such as mosquitoes, but has not been observed in *Drosophila*, where increases in cGMP-IR have been detected in Inka cells [36], but not CCAP neurons [52]. Furthermore, CCAP neurons (and CCAP itself) are not required for larval ecdysis in *Drosophila* [53], suggesting that other neurons (in addition to or instead of) CCAP neurons must express an EH receptor. Here we show that EHR is widely expressed in hundreds of neurons including CCAP neurons, as well as in the trachea and Inka cells. Why all these targets do not show increases in cGMP at ecdysis remains unknown, but may be due to the sensitivity of the anti-cGMP antibody [52].

Interestingly we found significant colocalization of EHR and ETHR within the CNS, revealing that EH and ETH not only coordinate each other's release, but share common targets in the control of ecdysis. In addition, EHR is expressed in the principal EH-producing Vm neurons themselves and is also co-expressed with receptors to neuropeptides that have historically been viewed as acting downstream of EH (such as bursicon and CCAP). Thus, these results indicate the presence of both feedback and feedforward loops in EH signaling. Coherent feedforward loops in which both signals coordinately activate a downstream target are often used to filter transient activation by a primary signal and delay the onset of full activation [54]. Such features might underlie, at least in part, the sequential release of neuropeptides during ecdysis and the serial activation of the different ecdysis phases. In addition, we find considerable convergence of EH and ETH signaling, including at the Vm and CCAP/bursicon neurons. This convergence suggests synergistic physiological effects of the two signaling systems on downstream neurons, perhaps to dynamically regulate excitability, though this remains to be determined. In any case, the architecture of control revealed here, with EH and ETH mutually reinforcing each other's release and coordinating each other's downstream actions, is reminiscent of the architecture of developmental gene regulatory networks, in which transcriptional determinants establish a regulatory state and then activate downstream inductive signals [55].

In addition to the CNS, we found that EHR was expressed in the epithelial cells of the trachea, including cells from the primary tubular branch, secondary branches, and terminal branches (Fig 4C and 4D), and that knockdown of EHR in trachea caused failures at ecdysis. These findings are consistent with previous reports that showed that ablating the Vm neurons [37] or disabling the *Eh* gene [28] caused defects in the vital process of tracheal air-filling that occurs during ecdysis. These defects cannot be rescued by injections of ETH [36] indicating that they are due to the action of EH and are not due to the absence of ETH release associated with these genotypes.

Although the identity of all EH targets largely remains to be determined, the relatively penetrant effects of EHR suppression in sets of peptidergic neurons, as opposed to neurons that use fast neurotransmitters (see Table 1), suggests that EH signaling is most important in modulating cells in the ecdysis network above the level of the synaptic networks that directly generate behavior. This is consistent with a model in which modulatory interactions regulate behavioral

state, which is then translated into the execution of state-specific motor programs. The fact that both ecdysis motor programs and the function of subsets of EHR-expressing cells vary with developmental stage suggests that either the identity or the importance of the downstream factors recruited by EH changes over development. This plasticity in ecdysis peptide usage is also seen phylogenetically, with the complement of essential ecdysial peptides varying across insect species. For example, whereas CCAP and orkokinin are dispensable for ecdysis in *Drosophila* [53,56], both are essential in the kissing bug, *Rhodnius prolixis* [57,58]. The stage-dependent differences in function of both ecdysial peptides and the cell types that express them in *Drosophila* may, in fact, simply reflect necessary differences in the control and generation of the ecdysis motor programs that evolved to accommodate metamorphosis and its attendant changes in body plan [59]. Overall, our results suggest an intricate interplay between multiple cell types and between multiple neuromodulatory peptides in regulating ecdysis. Understanding of how this complex behavior is controlled in *Drosophila* and, potentially, in other species, should contribute to our understanding of how neuromodulation mediated by neuropeptides regulates animal behaviors.

## Materials and methods

### Fly strains

Fly stocks were maintained at room temperature (22–25ºC) on standard cornmeal agar media under a 12h light/12h dark regime. Stocks used and their source are listed in S1 Table.

### *EHR*-GAL4 generation

The *EHR*-GAL4 line was made by inserting a Trojan GAL4 Expression Module (TGEM) [31] into the intron separating exons 10 and 11 of the CG10738 gene (S1 Fig). Insertion was targeted to the following site using CRISPR/Cas9:
   AAATGTGGGCTGTACTTATTAGG (PAM site underlined). To make the TGEM construct, homologous arms of 1kb flanking the Cas9 cleavage site were amplified by PCR using the following primer pairs synthesized by Integrated DNA Technologies, Inc. (Coralville, Iowa, USA):

EHR-3NotI: AGTCAG GCGGCCGC AAGTACAGCCCACATTTTGC,

EHR-3AgeI: AGTCAG ACCGGT TGATGGAAGTCCTGGATTCG,

EHR-5SpeI: AGTCAG ACTAGT AATAGGCATCCCCTCGTTGT,

EHR-5AscI: AGTCAG GGCGCGCC ATTAGGTTATTTTAAAGTGCATGCAGAAGG,

   The PCR products were cloned into the pT-GEM(0) vector. The resulting plasmid (250-pGS3-AgeI-NotI-arm-SA-T2AGal4-hsp70-3xP3-RFP-SV40-AscI-EHRarm) was co-injected with a pBS-U6-sgRNA-ETH plasmid encoding the guide RNA into embryos of flies expressing germline Cas9 as described previously [31]. Transformants were identified by their expression of the 3xP3-RFP marker. The EHR Split-GAL4 hemidriver lines were generated from the EHR-Gal4 strain by ΦC31-mediated cassette exchange as described in Diao *et al.,* (2015) [31].

### UAS-EHR generation

GC10738 cDNA was amplified from *Drosophila* larval CNSs using standard methods. The entire coding sequence was re-amplified using primers that included XhoI and KpnI sites at the 5' and 3' ends, respectively. Correct sequence was confirmed and corresponds to isoform called isoform "C" in Flybase (https://flybase.org/). The XhoI-KpnI fragment (containing the entire CG10738 coding sequence) was cloned into pUAST transformation vector by Genewiz (New Jersey, USA). Transformants were generated by BestGene (California, USA).

## EH synthesis

Synthetic EH was produced by GenScript (Hong Kong) using the pFastBac baculovirus expression system. Several constructs were tested with the goal of producing a secreted protein with high yield. Highest expression was obtained with a design that included the putative native signal sequence (vs. the signal sequence of the major envelope glycoprotein, gp67) and a His-tag at the amino-terminal of the protein. The sequence of the expressed protein was (putative mature EH sequence underlined):

MNCKPLILCTFVAVAMCLVHFGNAHHHHHHENLYFQGLPAISHYTHKRFDSMGGIDFVQVCLNNCVQCKTMLG-DYFQGQTCALSCLKFK GKAIPDCEDIASIAPFLNALE

Nevertheless, analysis of the medium indicated that the protein was not secreted and it was therefore purified from the cell lysate using an Ni-NTA column. Higher purity fractions were pooled and sterilized by 0.22 µm filter sterilization. Proteins were analyzed by SDS-PAGE and Western blot for molecular weight and purity measurements using standard protocols. The primary antibody for Western blot was a mouse-anti-His and the concentration was determined by Bradford protein assay with BSA as a standard. Whether the signal peptide was cleaved and the protein was correctly folded were not determined. The purified peptide was aliquoted in 50 mM Tris-HCl, 500 mM NaCl, 5% Glycerol, pH 8.0 at a concentration of 0.25 mg/ml and stored at -20°C.

## Stimulation with synthetic EH

Larvae were reared at 25°C on Petri dishes with fly medium. First instar larvae approaching ecdysis to the second instar were identified by their double mouth hooks (dMH) and placed individually on Petri dishes coated with apple juice agar. Once they first pigmented the plates of the second instar (double vertical plate stage, dVP, approximately 25 min prior ecdysis to the 2nd instar; [32]) they were monitored until they were ca. 10 minutes before the start of the ecdysial sequence (but showed no signs of tracheal collapse). Their tracheae were then dissected in PBS and incubated in Schneider's insect medium (Sigma). All tracheae of the same genotype dissected within 20 min were pooled; then were challenged either with synthetic EH (1nM final concentration) or the same volume of water. After 1h incubation at room temperature the tracheae were fixed and processed for ETH immunoreactivity [32], as described below. Tracheae from wild type animals prior to ecdysis as well as from larvae post-ecdysis were included as controls. In the case of animals mutant for EHR (which do not ecdyse) tracheae were dissected 2h after the dVP stage.

All Inka cells were imaged using the same parameters using an Olympus DSU spinning disk microscope (40x oil objective Olympus). The intensity of ETH immunoreactivity (ETH-IR) of each Inka cell was scored blind to both genotype and condition and assigned a score from 0 (no detectable ETH-IR) to 3 (maximal ETH-IR). Thirty to 60 cells were measured per conditions, representing the tracheae of 3–7 animals.

## Immunohistochemistry

Preparations were dissected on Sylgard-coated plates under cold PBS and fixed in 4% paraformaldehyde for 1h at room temperature or overnight at 4°C, as previously described in Clark, *et al.* (2004) [36]. Tissues were then rinsed several times in PBS with 0.3% Triton X-100 (Sigma) (PBSTX) and incubated in primary antibodies overnight at 4°C with constant agitation. Then they were washed in PBSTX and incubated in fluorescent secondary antibodies (Jackson ImmunoResearch, West Grove, PA) and mounted onto poly-L-lysine (Sigma) coated coverslips. Primary antibodies used include: rabbit anti-CCAP ( [52]; used at 1:500), rabbit anti-ETH ( [32] used at 1:2,000) and rabbit anti-EH ( [28]; used at 1:200). For GAL4 lines that produced low GFP signal when driving UAS-GFP, tissues were processed using anti-GFP (Invitrogen; used at 1:1,000). Preparations were viewed and imaged under a conventional fluorescent microscope or under an Olympus DSU spinning disk microscope and analyzed using ImageJ [60].

PLOS Genetics

### Survival across the different ecdyses

To quantify the number of animals that successfully completed larval ecdysis, we placed first instar larvae of the relevant genotype on Petri dishes with regular food and counted the number of larvae that ecdysed to the second instar. These larvae were transferred to Petri dishes with new regular food, and we then counted the number of animals that ecdysed to the third larval instar. We then transferred these larvae to vials with regular food, where we followed the success of pupal and adult ecdysis. For some genotypes we preferentially focused on specific ecdyses.

### Quantification of larval ecdysis behavior

Larvae were reared at 25ºC on Petri dishes with fly medium. First instar larvae at the dMH (double mouth hook) stage were placed on Petri dishes coated with apple juice agar with yeast. At dVP larvae were transferred individually to a new slightly wet agar Petri dish and video-recorded under a Leica dissecting microscope (Nussloch, Germany) until they completed ecdysis or for up to 2h after the time of ecdysis of control animals. Behavioral analysis scored for the presence or absence of three different phases: "locomotion," consisting of normal locomotor activity; "pre-ecdysis," consisting of anterior-posterior contractions (AP) and squeezing waves (SW); and "ecdysis" which started with "biting" behavior and ended with the final backward thrust, regardless of whether the old cuticle was eventually shed. "Atypical pre-ecdysis" was defined by partial, weak, or missing AP contractions or SW (as described previously in Scott, *et al.* (2020) [29]. Quantification of tracheal dynamics included: (1) the time taken to completely fill the trachea with air; for those animals that successfully completed that process and, (2) the percentage of animals that completed the air filling process (if ≥ 95% of the trachea filled with air it was considered filled) or failed.

### Quantification of pupal ecdysis and adult eclosion

Flies were reared at 25ºC on standard cornmeal agar media under a 12h light/12h dark regime. Pupal and adult ecdysis survival and failure were evaluated as described in Lahr, *et al.* (2012) [53]. For experiments using the TARGET system (temperature sensitive GAL80$^{ts}$; [35]) or the temperature-sensitive cation channel, *TrpA1* [61], flies were reared at 18ºC and transferred to 30ºC before pupal ecdysis or adult eclosion (24h hours before ecdysis for GAL80$^{ts}$, and 2h for *TrpA1* flies), and ecdysis success or failure was evaluated.

### RNAi screening

Six different UAS-RNAi lines for CG10738 (see S2 Table) were crossed with the *tubulin*-GAL4 line and the severity of the ecdysial defects determined by counting the number of larvae or pupae showing failures at ecdysis and the number of adults that emerged (see S2 Table). This screen revealed that line CG10738 RNAi-R1 produced the most severe phenotype and is the one used here; it was always used in combination with UAS-*dicer2* (UAS-*dcr2*) to boost the effectiveness of RNAi knockdown.

### Real-time GCaMP Imaging

In order to record calcium dynamics in *in vitro* CNS preparations, CNSs were dissected from animals around 2 hours prior to pupal ecdysis under cold 1X PBS and immediately mounted on a poly-L-lysine coated glass, placed in an Attofluor chamber (Thermo-Fisher Scientific), and covered with 2mL of Schneider's insect medium (Sigma). GCaMP6s [62] signal was recorded using an Olympus DSU spinning disk microscope with a 20X water objective and a Hamamatsu high sensitivity camera. Preparations were first imaged for 5min at 0.2 Hz, and preparations showing spontaneous activity were discarded. Synthetic ETH1 (600nM final concentration) or H$_2$O (control) was added and 5 Z-stacks across the whole VNC were recorded for one hour at 0.2 Hz, as previously described in Mena, *et al.* (2016) [25].

## Video analysis

Real-time calcium imaging videos were processed using FIJI software [60] to create hyperstacks and a maximum intensity projection (template). We then used the FluoroSNNAP MATLAB program to select all ROIs that corresponded to the center of each neuronal cell body and followed each of their activity during the 60 min recording period to create at the time series of fluorescence intensity. Calcium traces were normalized by using the previous 50 frames corresponding to 250 s as baseline, obtaining the dF/F traces (as described in [63] and [64]).

## Manual selection of cell types

We analyzed 5 different videos (n = 65.4 ± 10.78 cells; mean ± standard deviation), where we defined six EHR cell populations: posterior cells (POS, n = 10.2 ± 2.2), Left neurons (Left, n = 7.8 ± 0.40), Right neurons (Right, n = 6.8 ± 1.3), central cells Left (CEN L, n = 10.4 ± 2.4), central cells Right (CEN R, n = 10.2 ± 2.2) and the rest ("REST", n = 20.20 ± 13). These groups were defined by identity and their position within the VNC considering a midline that symmetrically separate left and right side of the CNS.

## Population coupling function

The functional population coupling function (PCF) of each cell was computed following Okun, *et al*. (2015) [46]. We computed the lagged-cross correlation function between each time series of one cell versus that of a 'population' time series, which was obtained by averaging the activity of all the remaining cells of the population. To then characterize the level and temporality of the coupling of each cell, the maximum of the function and its location in terms of temporal lags were extracted. Using this measure, a positive lag indicates that the population follows a given neuron, whereas a negative lag indicates that the neuron follows the population activity.

A linear-mixed effect model with experiments as random effects was used to evaluate if the lags associated with the maximum of the PCF were significantly different from zero. One model was used for each cell type. Means were estimated from these models, using the "REST" group as baseline. A Wilcoxon rank sums test was used to perform pairwise comparisons between the maximum value of the PCF, and the p-values were corrected by the Bonferroni method (15 comparisons).

## Statistics

Statistical analyses were performed using GraphPad Prism (GraphPad Software, La Jolla, CA, USA) and visualized using Adobe Illustrator. If data were normally distributed and had equal variance, they were analyzed by unpaired t test or one-way ANOVA followed by the Tukey test for *post hoc* multiple comparison analyses. Categorical data were analyzed by Fisher's exact test. The exact values for each comparison are shown in S3 Table.

## Supporting information

**S1 Fig. EHR protein and gene map.** Schematic representation of the CG10738 protein (A) and gene (B). (A) Protein symbolized as a black box includes three domains predicted by InterPro. (B) Gene map shows the location of insertions used here, which include the *EHR*-GAL4 (T-GEM construct) and a Minos insert. Non-coding regions are indicated as white boxes, coding regions as red boxes, and introns as black lines. Scale bar: 1 Kb.
(TIF)

**S2 Fig. Activation of all EHR-expressing cells using TrpA1 caused failures at pupal ecdysis and at adult eclosion.** Percentage of animals that failed to complete pupal ecdysis (A), and adult eclosion (B), after activation of EHR-expressing

cells using UAS-*TrpA1* for 2h before ecdysis. Statistical analyses were performed using Fisher's exact test, and significant differences are indicated by letters (see results in S3 Table).
(TIF)

**S3 Fig. EHR is broadly expressed in somatic tissues and across developmental stages.** Whole body pattern of EHR expression visualized with GFP (*EHR*>GFP) at the first (A), second (B), and third (C) larval instar; at the P3 pre-pupal stage (D); and at the pharate adult stage (E). Insert in (C) shows EHR expression in some cells of the body wall; white arrowheads in (E) point the EHR expression in the proboscis (top) and in dorsal bands of cells of the body wall (bottom). Scale bar 200 µm for panels and (A-E), and 100 µm for insert in (C).
(TIF)

**S4 Fig. Inka cells do not express the ETH receptor.** ETH receptor (ETHR) expression (white arrowhead) in third larval instar tracheal preparation visualized using a membrane-bound GFP (*ETHR*>mCD8-GFP, in green), together with ETH-immunoreactivity (in red)(white arrow). ETHR-expressing cell is immediately adjacent to the Inka cell. Tr: dorsal tracheal trunk.
(TIF)

**S1 Video. Real-time imaging of intracellular calcium signaling in EHR-expressing neurons.** (A) Example of time-course of *ex vivo* calcium activity in pre-pupal *EHR*>GCaMP6s CNS) following stimulation with synthetic ETH (600 nM). Images were captured for an hour at a frequency of 0.2 Hz (Video speed 120x). See Fig 8 for details.
(MP4)

**S1 Table. List of all the strains used and their source.**
(DOCX)

**S2 Table. EHR RNAi screening.** Phenotype of RNAi lines for EHR expressed using *tubulin*-GAL4.
(PDF)

**S3 Table: Summary of statistical analyses.**
(DOCX)

## Acknowledgments

We thank Eileen Krüger for amplifying CG10738 cDNA. We thank Vivian Budnik, Susan McNabb, Paul Taghert, Rob Jackson, Christian Wegener, and the Bloomington, Kyoto and Vienna, *Drosophila* stock centers for fly stocks, and Michael Adams for anti-ETH antisera.

## Author contributions

**Conceptualization:** Valeria Silva, Robert Scott, Paulina Guajardo, Ruben Herzog, Benjamin H. White, John Ewer.

**Data curation:** Valeria Silva, Robert Scott, Ruben Herzog, Benjamin H. White, John Ewer.

**Formal analysis:** Valeria Silva, Robert Scott, Ruben Herzog, Benjamin H. White, John Ewer.

**Funding acquisition:** Benjamin H. White, John Ewer.

**Investigation:** Valeria Silva, Robert Scott, Paulina Guajardo, Haojiang Luan, Ruben Herzog, John Ewer.

**Methodology:** Valeria Silva, Robert Scott, Ruben Herzog, Benjamin H. White, John Ewer.

**Project administration:** John Ewer.

**Resources:** Benjamin H. White, John Ewer.

**Supervision:** John Ewer.

**Validation:** Valeria Silva, Robert Scott, Haojiang Luan, Ruben Herzog, John Ewer.

**Visualization:** John Ewer.

**Writing – original draft:** Valeria Silva, Benjamin H. White, John Ewer.

**Writing – review & editing:** Valeria Silva, Robert Scott, Paulina Guajardo, Benjamin H. White, John Ewer.

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
