## [Decision Letter · Decision Letter 0]

15 May 2025

PGENETICS-D-25-00392

Characterization Eclosion Hormone Receptor function reveals differential hormonal control of ecdysis during Drosophila development.

PLOS Genetics

Dear Dr. Ewer,

Thank you for submitting your manuscript to PLOS Genetics. After careful consideration, we feel that it has merit but does not fully meet PLOS Genetics's publication criteria as it currently stands. Therefore, we invite you to submit a revised version of the manuscript that addresses the points raised during the review process.

Please submit your revised manuscript within 60 days Jul 14 2025 11:59PM. If you will need more time than this to complete your revisions, please reply to this message or contact the journal office at plosgenetics@plos.org. Please include the following items when submitting your revised manuscript:

We look forward to receiving your revised manuscript.

Kind regards,

Chun Han, Ph.D.

Academic Editor

PLOS Genetics

Fengwei Yu

Section Editor

PLOS Genetics

Aimée Dudley

Editor-in-Chief

PLOS Genetics

Anne Goriely

Editor-in-Chief

PLOS Genetics

**Journal Requirements:**

Potential Copyright Issues:

i) Please confirm (a) that you are the photographer of 2B, and 3B, or (b) provide written permission from the photographer to publish the photo(s) under our CC BY 4.0 license.

5) Thank you for stating "All data and results are included in the submission." If your research concerns only data provided within your submission, please write "All data are in the manuscript and/or supporting information files" as your Data Availability Statement.

**Reviewers' comments:**

Reviewer's Responses to Questions

Reviewer #1: This is an exceptionally clear and incisive report on the neuropeptide EH receptor in Drosophila. EH (eclosion hormone) is part of a complex system underlying neuropeptide-regulated behavior, that features hierarchical control levels, and mutual positive feedback between the two key excitatory neuropeptides at the top of the hierarchy (ETH and EH). The regulated behavior is ecdysis, which represents the final step in the molting process of insects whereby the newly molted animal breaks out of its old cuticle through a set of stereotyped behaviors. The behavioral sequence promotes extraction of the body and appendages without damage and occurs at the end of each molt; embryonic and adult ecdysis are termed eclosion.

The EHR had been a poorly understood element in the system. This report provides much-needed information on its expression and functional utilization in diverse target cells. Unlike the majority of neuropeptide receptors, EHR is a membrane guanylate cyclase which adds novelty to the report. Other strengths found in the paper – they perform extensive mosaic analysis using split Gal4s to map the identities of the many EHR-expressing cells. They also perform mosaic EHR expression to relate different target cells to different stage-specific behaviors. They also document Ca dynamics in EHR cells of the larval CNS following exposure to EH. In all, this is an exciting, technically-proficient and highly-informative paper.

Minor concerns

Figure 1. The intensity of anti-ETH immunoreactivity is quantified as a function of time in Inka cells. Quantification is presented as arbitrary subjective categories. Is it possible to measure fluorescence intensity?

Figure 2. In the HER mutant state, the locomotion that precedes the TC stage appears to start early and its duration appears longer due to the premature onset. Alternatively, could it start at the correct time but extend in duration because the TC stage is delayed? Are there independent markers that suggest whether the behavior of the TC is anomolously displayed?

Figure 4A. Suggest placing the pink Inka cells in the schematic on the side of the trachea, for better visibility.

Figure 4 B/C. Why are the majority of EHR-nls RFP cells not labeled by EHR> mCD8? Only the Inka cells appear to be mCD8-+. Also, the Legend says the Inka cells is readily visible with nuclear reporter in 1C – but that is not visible to me.

Figure 5. EHR in larval CNS at the 3rd instar. How does this compare to the 1st instar CNS? Is there an increase in cell number? Are there differences in ecdysis behavior between the one at 1 to 2, versus 3 to Pupal?

Figure 6. EHR expression in the pharate adult CNS. Is this pattern retained in the adult? Or do these cells die are their use at adult eclosion?

Figure 8. The major lines are marked by dark versus light grey – these are very similar and I suggest you try something else.

Given expression of EHR in Vm cells it may be worth noting whether ETH-R is expressed by Inka cells.

Is EHR also expressed by other EH-expressing neurons (non-Vm)?

Also, EH-Gal4 is not marked when used.

Also UAS-EHR is only marked when used in the case of EH-Gal4, but not when used with other Gal4s.

Reviewer #2: EH is an important regulatory neuropeptide involved in insect molting or ecdysis. The EH receptor (EHR) has been identified in a few other insect species, including the oriental fruit fly and the desert locust; however, its function in Drosophila melanogaster, a model insect for ecdysis research, remains unexplored. In this study, the authors investigated the role of EHR in fruit fly ecdysis using newly developed molecular genetic tools, including EHR mutant and Gal4 lines. Comprehensive molecular genetic analyses using targeted knockdown and genetic rescue experiments revealed that the EHR's role varies depending on the developmental stage and target. This work is an important contribution to resolving the long-standing uncertainty about the neuronal functions of EH in Drosophila ecdysis. However, Figure 7 is not sufficiently developed and, in its current form, is not relevant to the functions of EHR that are the focus of this manuscript.

Major comments,

In Figure 7, the authors showed that ETH stimulates neural activity patterns in neurons that express EHR (most of which also express ETHR, such as CCAP neurons). The observation that ETH induces neural activity patterns in these neurons has been reported several times, both by the authors' group and by other groups. The actual new important finding should be to investigate the role of EHR in regulating the neural activity of these neurons stimulated by ETH. In addition, co-treatment of the CNS with EH could induce changes in the fictive ecdysis patterns induced by ETH, allowing the authors to investigate the function of EHR in such changes. This is particularly important because the authors claim that EH and ETH acts synergictially in the brain neurons.

The authors also discovered an early activation of CCAP cells in ETH-induced fictive pupal ecdysis, compared to other EHR neurons. Since CCAP cells are also ETHR cells, the activation of CCAP cells during pupal ecdysis is expected to be directly induced by ETHR rather than EHR. Without showing the role of EHR in such CCAP cell function, the relevance of this result to this work is limited. I suggest to remove it.

EHR in CCAP neurons rescued larval ecdysis EHR mutants to some extent (Table 2). This does not fit well with the previous observation that CCAP neurons are not required for larval ecdysis in Drosophila. Does this require some explanation?

Minor points

1. Authors claim that EHR is a bona fide receptor for EH. Thus, it would be useful to include EH mutants at least in Table 2.

2. 64 lines. “that that”

3. Fig. 2B, 3B. Include enlarged image of the mouth part.

4. Fig. 2A, 2C, 2E, 3A, 3C-F, 8A-D. Include statistical comparisons.

5. Fig. 4F. What about other imaginal discs?

6. Table 1. What does 'ok' mean? Do all 'ok's indicate ok^3? If so, why do some genotypes have ‘ok’ labels in larval and pupal ecdysis, while others have it only in pupal ecdysis? What does ‘-’ indicate?

Reviewer #3: This paper tests the hypothesis that the Drosophila gene CG10738, a homolog of the eclosion hormone receptor-expressing gene previously described in the oriental fruit fly Bactrocera dorsalis, encodes the receptor for eclosion hormone (EHR). Through a series of elegant experiments, the authors provide very convincing evidence that this is the case. They have shown that cells expressing EHR respond to synthetic eclosion hormone (EH), that EHR-deficiency causes lethal ecdysis phenotypes in all stages of development (larval, pupal, adult), and that expression of this receptor in EHR mutant cells rescues lethal phenotypes. Of great interest are findings that EHR and ecdysis-triggering hormone (ETH) receptors are co-expressed in cells that produce these signaling molecules (Inka cells and Vm neurons) and in many “downstream” target cells known to be involved in orchestrating the ecdysis behavioral sequence. In addition to its role in coordinating ecdysis behaviors, results support key functional roles for EHR in tracheal airfilling, wing expansion, cibarial pumping, location in leg imaginal discs associated with Keilin’s organ and in other adult structures. EHR-expressing cells are mapped in the central nervous system and patterns of cellular activity during fictive pupal ecdysis have been monitored via calcium imaging.

This study of eclosion hormone functions is a tour de force that demonstrates how a behavioral sequence is programmed by a feedback and feed-forward modulatory signaling network. The work builds upon and refines previous studies by the authors and others, and greatly expands our understanding of how a complex behavior is initiated and scheduled.

Specific comments

1. There is very little to criticize about this work. A few suggestions are offered in the interests of promoting clarity.

2. Synthetic eclosion hormone. How was it purified from the cell lysate? Presumably nickel resin. Were the signal sequence and His-tag portions cleaved from the mature peptide? If not, it is interesting that the entire peptide retained activity. Did disulfide bond formation occur prior to cell lysis? Within the Sf9-expressing cells? How was concentration determined?

3. Fig. 2. Please explain in more detail the terms “locomotion pre TC” and “locomotion after TC” in Fig. 2A. Although this is somewhat explained in the Methods, perhaps it could be included in the figure legend. Are they simply normal crawling movements?

4. The cells that produce ETH are variously referred to as Inka cells, ETH cells, epitracheal cells. It would be helpful to settle upon a consistent name (Inka cells?), especially since they are not a heterogenous population of cells.

5. The three sentences beginning with line 177 could be re-ordered or re-formulated. It is stated that premature activation of EHR-expressing cells causes 100% lethality and therefore these cells are essential for ecdysis success. The third sentence alludes to the necessity of correct timing, but this may not be so clear to the uninitiated reader.

6. Line 214. Control of ecdysis by EH through both feed-forward and feedback mechanisms fits with the previous work of Diao et al. (Ref 17) on neuromodulatory connectivity. It is surprising that this is not brought out in the Discussion section.

7. Lines 339-341: This sentence states that because small subsets of EHR-expressing neurons failed to rescue ecdysial defects, it is concluded that many EH targets are important for ecdysis success. The logic here is difficult to discern.

8. Line 500: Please define dMH stage of 1st instar larvae (spell it out for the non-initiated).

9. Line 538: Was this Drosophila ETH1?

10. Regarding the Supplementary video, is it not possible to label/identify at least some of the EHR neurons responding to ETH?

**Have all data underlying the figures and results presented in the manuscript been provided?**

Reviewer #1: Yes

Reviewer #2: **No: ** There is no spreadsheet

Reviewer #3: Yes

PLOS authors have the option to publish the peer review history of their article (what does this mean? ). If published, this will include your full peer review and any attached files.

**Do you want your identity to be public for this peer review?** For information about this choice, including consent withdrawal, please see our Privacy Policy .

Reviewer #1: No

Reviewer #2: No

Reviewer #3: **Yes: ** Michael E. Adams

**Figure resubmission:**
---

## [Decision Letter · Decision Letter 1]

27 Jul 2025

PGENETICS-D-25-00392R1

Characterization of Eclosion Hormone Receptor function reveals differential hormonal control of ecdysis during Drosophila development.

PLOS Genetics

Dear Dr. Ewer,

Thank you for submitting your manuscript to PLOS Genetics. After careful consideration, we feel that it has merit but does not fully meet PLOS Genetics's publication criteria as it currently stands. Therefore, we invite you to submit a revised version of the manuscript that addresses the points raised during the review process.

In your revised manuscript, please address all minor concerns of the reviewers. Addressing the concern of Reviewer #2 on the synergistic effects of ETH and EH is welcomed but not required.

Please submit your revised manuscript within 30 days Aug 26 2025 11:59PM. If you will need more time than this to complete your revisions, please reply to this message or contact the journal office at plosgenetics@plos.org. Please include the following items when submitting your revised manuscript:

We look forward to receiving your revised manuscript.

Kind regards,

Chun Han, Ph.D.

Academic Editor

PLOS Genetics

Fengwei Yu

Section Editor

PLOS Genetics

Aimée Dudley

Editor-in-Chief

PLOS Genetics

Anne Goriely

Editor-in-Chief

PLOS Genetics

**Journal Requirements:**

**Reviewers' comments:**

Reviewer's Responses to Questions

**Comments to the Authors:**

Reviewer #1: The authors did an excellent job in responding to reviewer’s concerns and questions, including mine. With a second opportunity to review the findings, I had a couple of additional questions for minor concerns.

First on a technical note, I see that Flybase predicts two different protein isoforms for CG10738 (EHR). They differ by about 15 amino acids in the middle of the protein. Is there any information about this from prior studies of EHR? Or from other guanylate cyclases? Which form (PC/PD or PE/PF) was used in the rescue experiments? The methods section does not include that detail.

Second, regarding the diversity and functional heterogeneity of different EHR-expressing cell types (Tables 1 and 2): I was interested to see that nsyb Gal4 KD of EHR produced strong effects in pupal but not larval ecdysis. A very similar pattern was found with btl-Gal4. But remarkably, btl-Gal4 could drive strong rescue of HER mutants in larval stages stages, but not in pupae. Do the authors have any speculation as to why certain EHR bearing cells (btl-expressing) might be sufficient, while not required, for EHR function at some stages? Compensatory functions for btl and nsyb populations? Was the rescue also tried with n-sybGal4?

Reviewer #2: Major comments:

In response to my previous (major) comment regarding Figure 7 (now Figure 8), the authors explained that treating the brain with both ETH and EH during the in vitro imaging experiments is technically infeasible. However, I believe the authors can evaluate the synergistic effects of ETH and EH on CCAP neurons by investigating Ca²⁺ activity in CCAP cells expressing EH-R-RNAi or ETH-R-RNAi during ETH-induced fictive ecdysis behavior.

Minor comments:

1. The knockdown efficiency of EH-R-RNAi was not evaluated. I believe it is necessary to evaluate the EH-R knockdown efficiency using EH-R>EH-R-RNAi in the brain or CNS.

2. Figure 1: "F" and "G" should be labeled "E" and "F," respectively.

3. Supplemental Fig. 1 shows mimic cassette insertion, which was not used or mentioned in the manuscript.

4. Line 220: "C, F" should be "C-F."

Reviewer #3: The authors have responded to all of my questions and suggestions very well. This includes providing information about production and quantification of eclosion hormone and clarification of various other issues. A couple of issues remain:

1. One was intrigued by the authors’ response to Reviewer 2 regarding the narrow window of Inka cell responsiveness to EH. The authors state that “Inka cells did not respond to EH unless they were challenged less than 10 minutes before ecdysis (results shown in Fig. 1)”. In fact, exposure to EH in this figure occurs only at a single time point: dVP (double vertical plates). According to Park et al., (2002); Ref 32, dVP occurs 10 min prior to tracheal collapse (TC), 15 min before pre-ecdysis, and ~25 min prior to ecdysis. Are the authors stating that Inka cells are insensitive to EH prior to dVP? Although Fig. 1 does not present any data in support of this claim, it is a very important point. Would it be possible to add some negative data on Inka cell responsiveness prior to dVP in Fig. 1 to solidify this finding?

2. Regarding the figure legend of the revised Supplementary Video 1, reference to Fig 7 should now be to Fig 8, since two figures have been exchanged in the revised manuscript.

**Have all data underlying the figures and results presented in the manuscript been provided?**

Reviewer #1: Yes

Reviewer #2: Yes

Reviewer #3: Yes

PLOS authors have the option to publish the peer review history of their article (what does this mean? ). If published, this will include your full peer review and any attached files.

**Do you want your identity to be public for this peer review?** For information about this choice, including consent withdrawal, please see our Privacy Policy .

Reviewer #1: **Yes: ** Paul Taghert

Reviewer #2: No

Reviewer #3: **Yes: ** Michael Adams

**Figure resubmission:**
---

## [Editor Report · Decision Letter 2]

6 Aug 2025

Dear Dr Ewer,

We are pleased to inform you that your manuscript entitled "Characterization of Eclosion Hormone Receptor function reveals differential hormonal control of ecdysis during Drosophila development." has been editorially accepted for publication in PLOS Genetics. Congratulations!

Yours sincerely,

Chun Han, Ph.D.

Academic Editor

PLOS Genetics

Fengwei Yu

Section Editor

PLOS Genetics

Aimée Dudley

Editor-in-Chief

PLOS Genetics

Anne Goriely

Editor-in-Chief

PLOS Genetics

Comments from the reviewers (if applicable):

**Data Deposition**

http://datadryad.org/submit?journalID=pgenetics&manu=PGENETICS-D-25-00392R2

**Press Queries**

---

## [Editor Report · Acceptance letter]

PGENETICS-D-25-00392R2

Characterization of Eclosion Hormone Receptor function reveals differential hormonal control of ecdysis during Drosophila development.

Dear Dr Ewer,

We are pleased to inform you that your manuscript entitled "Characterization of Eclosion Hormone Receptor function reveals differential hormonal control of ecdysis during Drosophila development." has been formally accepted for publication in PLOS Genetics! Your manuscript is now with our production department and you will be notified of the publication date in due course.

With kind regards,

Anita Estes

PLOS Genetics

On behalf of:
